# SNX8 enables lysosome reformation and reverses lysosomal storage disorder

Xinran Li [1,2,3,4,9], Cong Xiang [1,2,4,9], Shilei Zhu[1,2,4], Jiansheng Guo[5], Chang Liu[1,2,4], Ailian Wang[1,2,4], Jin Cao[1,2,4], Yan Lu[1,4,6,7], Dante Neculai [6,7], Pinglong Xu [1,2,3,4] & Xin-Hua Feng [1,2,4,8] ✉

Lysosomal Storage Disorders (LSDs), which share common phenotypes, including enlarged lysosomes and defective lysosomal storage, are caused by mutations in lysosome-related genes. Although gene therapies and enzyme replacement therapies have been explored, there are currently no effective routine therapies against LSDs. During lysosome reformation, which occurs when the functional lysosome pool is reduced, lysosomal lipids and proteins are recycled to restore lysosome functions. Here we report that the sorting nexin protein SNX8 promotes lysosome tubulation, a process that is required for lysosome reformation, and that loss of SNX8 leads to phenotypes characteristic of LSDs in human cells. SNX8 overexpression rescued features of LSDs in cells, and AAV-based delivery of SNX8 to the brain rescued LSD phenotypes in mice. Importantly, by screening a natural compound library, we identified three small molecules that enhanced SNX8–lysosome binding and reversed LSD phenotypes in human cells and in mice. Altogether, our results provide a potential solution for the treatment of LSDs.

Lysosomal storage disorders (LSDs) are devastating metabolic diseases caused by mutations in lysosome-related genes coding for hydrolases, transporters, ion channels, and trafficking proteins. More than 60 different types of LSDs are currently identified, and the summed occurrence is estimated to be between 1/5,000 and 1/5,500[1]. LSDs are featured by storing substances in lysosomes, leading to secondary lysosome-associated dysfunctions shared among LSDs[2]. For example, mutations in α-galactosidase A (GLA, Fabry Disease), β-hexosaminidase (HEXA/HEXB, GM2 gangliosidosis) or NPC1 (Niemann-Pick Disease type C1) will cause primary storage of globotriaosylceramide, GM2 ganglioside, or cholesterol, respectively[1]. However, they substantially share other lysosomal phenotypes such as enlarged lysosomes, reduced digestion, defective membrane trafficking/positioning, and impaired lysosome reformation[3]. The severity of LSDs depends on the nature of the causal mutation, but even mild LSD subtypes can result in death in youth, while severe subtypes can cause disease onset in infancy and death in childhood[1].

As genetic diseases, LSDs cannot be fully cured unless the causal gene is corrected, which is yet clinically unavailable. Currently, enzyme replacement therapies (ERTs), gene therapies, and substrate reduction therapies remain the few approaches established or under clinical trials, but each with severe limitations or drawbacks[1,2,4]. A common fault of all the above therapies is that they are specific to particular subtypes of LSDs. Due to the very low occurrence for each type of LSD, a broad-spectrum therapy for most, if not all, LSDs is desired.

[1]The MOE Key Laboratory of Biosystems Homeostasis & Protection and Zhejiang Provincial Key Laboratory of Cancer Molecular Cell Biology, Life Sciences Institute, Zhejiang University, 310058 Hangzhou, Zhejiang, China. [2]Center for Life Sciences, Shaoxing Institute, Zhejiang University, 321000 Shaoxing, Zhejiang, China. [3]ZJU-Hangzhou Global Scientific and Technological Innovation Center, 311200 Hangzhou, Zhejiang, China. [4]Cancer Center, Zhejiang University, 310058 Hangzhou, Zhejiang, China. [5]Center of Cryo-Electron Microscopy, Zhejiang University, Hangzhou, Zhejiang, China. [6]International Institutes of Medicine, The Fourth Affiliated Hospital of Zhejiang University School of Medicine, Yiwu, China. [7]Department of Cell Biology, and Department of General Surgery of Sir Run Shaw Hospital, Zhejiang University School of Medicine, Hangzhou, China. [8]The Second Affiliated Hospital, Zhejiang University, Hangzhou, Zhejiang, China. [9]These authors contributed equally: Xinran Li, Cong Xiang. ✉e-mail: fenglab@zju.edu.cn

Recent studies have shown that enhancing lysosomal functions globally may be a promising approach for broad-spectrum LSD therapies, considering that most LSD phenotypes are caused by secondary lysosomal dysfunctions rather than the primary gene defect[5,6]. Two significant pathways exist for the upregulation of lysosomal functions. One is the transcription of lysosomal genes through TFEB activation[7], and the other is the activation of lysosome reformation in response to a reduced number of functional lysosomes[8,9]. TFEB activation is a powerful way to boost lysosomal functions. Several reports have shown that activating TFEB-mediated transcription through overexpression of TFEB or molecular TFEB activators reduces LSD or neurodegeneration disease (ND) phenotypes[10-12]. However, as TFEB activation leads to upregulated expression of most lysosomal genes, its effect is non-specific and may lead to cellular stress[13].

On the other hand, lysosome reformation is an on-demand process that takes place on consumed lysosomes (e.g., autolysosomes, phagolysosomes, etc.)[8,9,14,15]. Thus, targeting lysosome reformation can be a more specific strategy for LSD therapies. As LSD cells often exhibit reduced lysosomal functions and impaired lysosome reformation[16-18], boosting lysosome reformation should restore functional lysosomes and reduce cellular damage. Many components of the lysosome reformation pathway were identified in the past decade, including mTOR and PIP5K1B for the initiation, kinesin-1, dynein and dynactin for the driving force, and clathrin, PIP5K1A among others for the membrane budding and the generation of proto-lysosomes[13,19,20]. Because the tubular structure serves as the platform for lysosome reformation, tubule formation is critical for the whole reformation process[8,19,21]. Yet, the factor(s) responsible for the tubule formation and maintenance remain unidentified.

Sorting nexins (SNXs) are a family of membrane-binding proteins featuring a phosphoinositide-binding PX domain[22]. 12 members of the SNX family carry a BAR domain[23] capable of inducing membrane curvature and thus are bestowed with tubule formation abilities. Indeed, multiple SNX-BAR proteins are found to mediate endosomal tubule formation and participate in endosomal retrograde transport[24-27]. Here we report that lysosomal tubule formation requires the function of SNX8 and, to a lesser extent, SNX2 - two members of the SNX-BAR family. The loss of SNX8 or SNX2/SNX8 induces LSD phenotypes, while overexpression of SNX8 can rescue LSD phenotypes in LSD model cell lines and a mouse model. Besides, we have also identified several small molecules that can reduce LSD phenotypes through up-regulating SNX8. Our results may give rise to potential broad-spectrum therapies for LSD patients.

## Results

### Sorting nexin 2/8 participate in lysosome tubulation

To examine whether SNX-BAR proteins participate in lysosome tubulation, we first checked their co-localization with LAMP1, a well-established lysosome marker. Our results showed that when overexpressed, only SNX2 and SNX8 were significantly co-localized with LAMP1, with SNX8 being best co-localized (Fig. 1A, B and Supplementary Fig. 1A, B). Immunostaining images confirmed that endogenous SNX2 and SNX8 were partially co-localized with endogenous LAMP1 (Fig. 1C). Live imaging analysis further showed that both SNX2 and SNX8 were localized onto tubular lysosomal structures under prolonged starvation (Fig. 1D, Supplementary Fig. 1C), under which condition lysosome reformation was reported to be triggered[9]. Therefore, SNX2 and SNX8 are candidate structural proteins for lysosome tubulation. It is worth noting, however, that some sorting nexins heterodimerize, and therefore may require the co-expression of the binding partner to function properly, thus our screen does not entirely rule out possible participation of other sorting nexins in lysosome tubulation.

To investigate if SNX2 and SNX8 are required for lysosome tubulation, we generated SNX2-knockout (KO), SNX8-KO, and SNX2/SNX8-double KO (DKO) HeLa cell lines (Supplementary Fig. 1D, E). Live

imaging experiments showed that lysosome tubulation was significantly reduced in SNX8-KO cells (Fig. 1E, F). SNX2-KO cells showed an insignificant reduction in lysosome tubulation compared to WT cells, while SNX2/SNX8-DKO further suppressed lysosome tubulation compared to SNX8 single KO (Fig. 1E, F). SNX8 overexpression rescued lysosome tubulation than SNX8-KO and SNX2/SNX8-DKO cells, confirming that this phenotype is not off-target (Figs. 1E, F). These data suggest that SNX8 may function as the major protein responsible for lysosome tubulation, while SNX2 only partially substitutes SNX8's function when the latter is absent.

### SNX8 binds to and tubulates lysosomes

To further examine the role of SNX2/SNX8 in lysosomal functions, we quantified the volume of lysosomes in SNX2 and/or SNX8-KO cells using lysotracker. SNX8 deficiency significantly increased lysosome volumes, probably due to enlarged lysosomes as judged by both images and quantifications (Fig. 2A–C, Supplementary Fig. 2A). SNX2/SNX8-DKO led to further increased lysosome volumes. Since suppression of lysosome reformation was reported to cause enlarged lysosomes, especially after overly sustained starvation[9,28], we monitored changes in lysosome volumes during starvation in these cells. In WT HeLa cells, lysosome volumes significantly increased in the first hour and then reduced at 6 h of starvation due to lysosome reformation[9] (Supplementary Fig. 2B, C). However, in SNX8-KO and SNX2/SNX8-DKO cells, lysosome volumes failed to reduce at 6 h of starvation (Supplementary Fig. 2B, C). Thus, failure of lysosome reformation in SNX8-KO and SNX2/SNX8-DKO cells caused enlarged lysosomes.

Next, we investigated the mechanism by which SNX8 regulates lysosome tubulation. Through co-immunoprecipitation (co-IP) experiments using an anti-LAMP1 antibody and in vitro binding assay using purified lysosomes (Supplementary Fig. 2D) and purified SNX8 proteins, we confirmed that SNX8 interacted with LAMP1, or at least LAMP1-positive lysosomes (Fig. 2D, Supplementary Fig. 2E). SNX proteins are known to bind phosphoinositides for their membrane targeting. PIP Strip assays showed that SNX8 could bind PI(5)P and PI(3,5)$P_2$, the phosphoinositides enriched on lysosomes[29]. In contrast, mutation of K135, the key phosphoinositide binding residue predicted by structural similarity (Uniprot, protein ID: Q9Y5X2) to known phosphoinositide-binding residues in other SNX-PX domains[30], to Ala (K135A) abolished PI(5)P binding and significantly diminished PI(3,5)$P_2$ binding (Fig. 2E). Therefore, SNX8 can be recruited to lysosomes through the interaction with PI(5)P or PI(3,5)$P_2$.

Interestingly, although PI(3,5)$P_2$ binding ability is impaired for SNX8-K135A, this mutant was still able to restore lysosome size at a lower efficiency than the WT SNX8 (Supplementary Fig. 2F). This is likely due to the high global concentration of the overexpressed protein, which waived the requirement of membrane targeting to create a high local concentration. In accordance, in cells treated with apilimod, a potent PIKfyve inhibitor that inhibits PI(3,5)$P_2$ synthesis[31,32], endogenous SNX8 shifts away from LAMP1-positive vesicles, but overexpressed SNX8 was still able to localize to lysosomes and suppressed the enlargement of lysosomes induced by apilimod (Supplementary Fig. 2G). When applied to purified lysosomes at 10 μM, SNX8 induced the formation of smooth tubules of diameters between 50-100 nm (Fig. 2F), while SNX8-K135A could form only thinner (30−50 nm diameter) tubular structures of distorted shapes (Fig. 2F).

### SNX2/8 deficiency causes LSD-like phenotypes

Lysosome reformation was shown to be critical for the homeostasis of functional lysosomes[9]. Disruption of lysosome reformation was shown to couple with lysosomal storage disorder phenotypes[17,33,34]. As tubulation is required for lysosome reformation, we examined LSD phenotypes in SNX8-KO and SNX2/8-DKO cells. Results showed that, in addition to enlarged lysosomes (Supplementary Fig. 2), SNX8-KO and SNX2/8-DKO cells also showed other LSD phenotypes, including

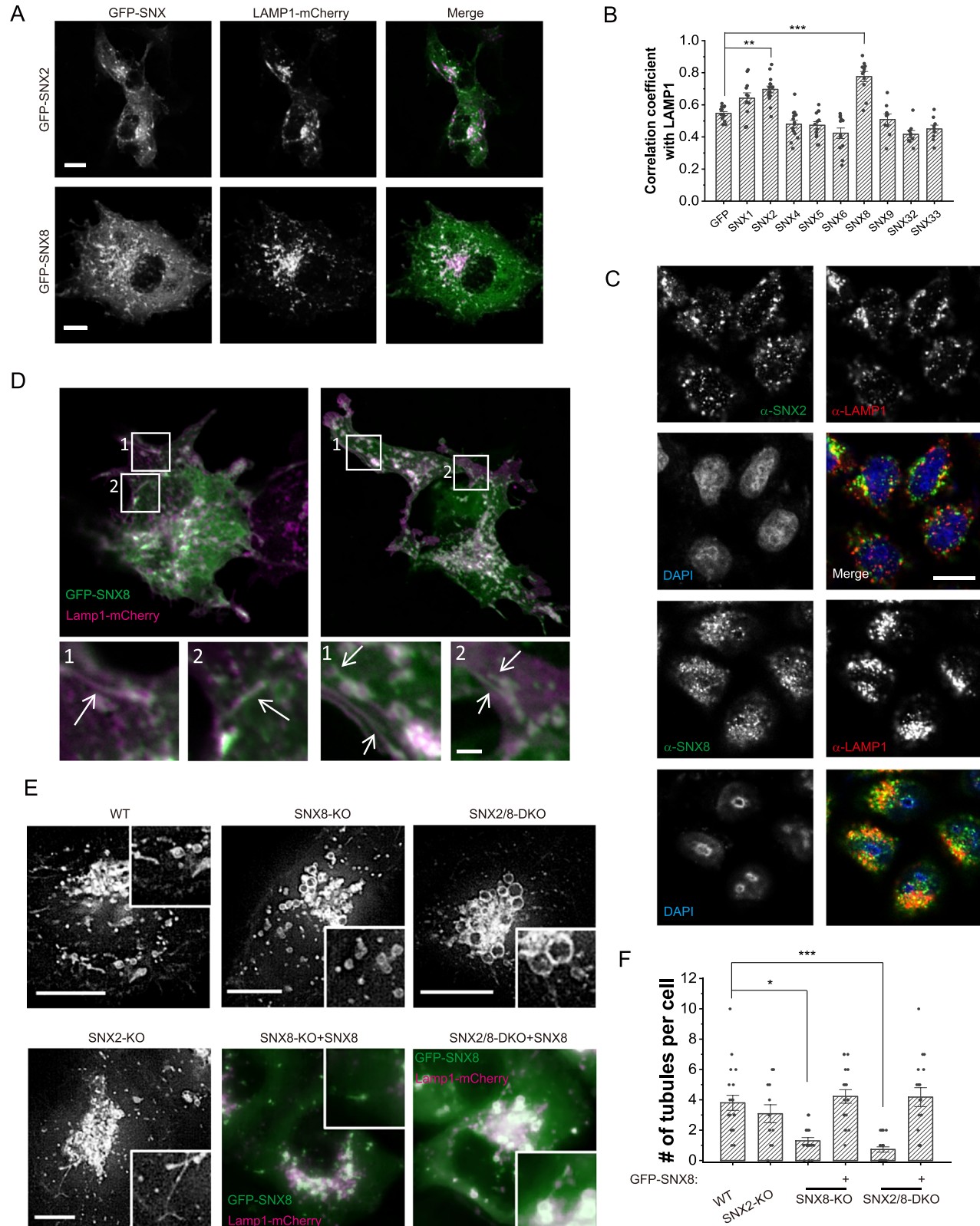

increased LAMP1 expression (Fig. 3A and Supplementary Fig. 3A), cholesterol accumulation (indicated by filipin staining, Fig. 3B, C), and defective lysosome-to-Golgi transport (demonstrated by pulse-chase experiments using BODIPY-Lactosylceramide) (Fig. 3D and Supplementary Fig. 3B). SNX8-KO and SNX2/8-DKO cells also showed increased cell death in response to severe starvation or repeated starvation (Fig. 3E, F, and Supplementary Fig. 3C, D). These results

suggest that impaired lysosome tubulation caused by the loss of SNX8 can lead to LSD phenotypes.

## Overexpression of SNX8 rescues LSD phenotypes in LSD model cells

In LSDs, most cellular phenotypes are caused by impaired lysosomal membrane trafficking and reduced lysosomal activity, which are more

**Fig. 1 | Sorting nexin2/8 participate in lysosome tubulation. A** SNX2 and SNX8 co-localize with LAMP1 under 24 h overexpression in COS1 cells. Scale bars = 10 μm. **B** Quantification of the correlation coefficient between SNX-BAR family members and LAMP1, with GFP as a negative control. Only SNX2 and SNX8 showed significant positive correlation with LAMP1 ($n$ = 13, 12, 15, 14, 13, 12, 11, 9, 9, 9, respectively, for each group, $p$ = 0.0036 (GFP vs. SNX2) and <0.00001 (GFP vs. SNX8)). **C** Immunofluorescence images of endogenous SNX2/SNX8 and LAMP1 in HeLa cells. SNX8 appeared to co-localize better with LAMP1. Scale bars = 10 μm. **D** COS1 cells were transfected with LAMP1-mCherry and GFP-SNX2 or GFP-SNX8 for 24 h, then serum-starved for 6 h before being subjected to live imaging. Insets show lysosomal tubules (white arrows). Scale bars = 1 μm. **E** WT, SNX2-KO, SNX8-KO, and SNX2/8-DKO HeLa cells were transfected with LAMP1-mCherry (and with GFP-SNX8 for SNX8-KO and SNX2/8-DKO cells) for 48 h, then serum starved for 16 h and monitored under confocal microscope. Scale bars = 10 μm. **F** Quantification of the average number of tubules per cell observed over a 1 min live imaging (frame rate = 0.5 fps) in WT, SNX2-KO, SNX8-KO, SNX8-KO + GFP-SNX8, SNX2/8-DKO and SNX2/8-DKO + GFP-SNX8 HeLa cells ($n$ = 20, 12, 17, 18, 22, 17, respectively, for each group, $p$ = 0.046 (WT vs. SNX8-KO) and <0.00001 (WT vs. SNX2/8-DKO)). For graphs, error bars are s.e.m, statistical comparison was done using one-way ANOVA, Tukey test (two sided, no adjustments). Source data are provided as a Source Data file.

often secondary defects than the primary defect from the causal mutation[1]. Boosting lysosomal activity and membrane trafficking can, therefore, theoretically rescue most of the cellular LSD phenotypes. Lysosome reformation facilitates lysosomal activity and recycling of the lysosomal membranes[9,35]. Since SNX8 is required for lysosome reformation, we investigated whether overexpression of SNX8 can rescue LSD phenotypes. For this purpose, we generated GLA-KO, HEXA-KO, and NPC1-KO HeLa cell lines (Supplementary Fig. 4A), modeling for Fabry Disease, GM2 gangliosidosis, and Niemann-Pick disease type C, respectively[1]. These cell lines all showed LSD phenotypes including enlarged lysosomes (Fig. 4A, B, Supplementary Fig. 4B), lysosomal cholesterol accumulation (Fig. 4C, quantified in Supplementary Fig. 4C), defective retrograde transport (Fig. 4D, quantified in Supplementary Fig. 4D), reduced lysosome tubulation (Fig. 4E, Supplementary Fig. 4E), and lower viability under repeated starvation (Fig. 4F), although the severity of the defect varies among them. Overexpression of SNX8 was able to rescue entirely or significantly reduce LSD phenotypes in all three cell lines (Fig. 4 and Supplementary Fig. 4B–E). As the three LSD cell lines differ in their causal factors of the disease, these results suggest that enhancing lysosome reformation through SNX8 overexpression was effective in rescuing LSD phenotypes in multiple types of LSD cells. Thus, based on *in cellulo* experiments, SNX8 may serve as a promising target for broad-spectrum LSD therapies.

### SNX8 overexpression rescues LSD phenotypes in *Hexb*[−/−] mice

To assess the potential of up-regulation of SNX8 as a therapeutic method for LSD, we utilized the *Hexb*[−/−] mice, a model for human Sandhoff disease[1]. Dermal fibroblasts isolated from neonatal *Hexb*[−/−] mice showed similar LSD phenotypes as HEXA-KO HeLa cells, including enlarged lysosomes (Fig. 5A, Supplementary Fig. 4F), cholesterol storage (Fig. 5B) and impaired lysosome tubulation (Fig. 5C). Overexpression of SNX8 significantly suppressed LSD phenotypes in these *Hexb*[−/−] fibroblasts (Fig. 5A–C, Supplementary Fig. 4F).

LSD patients usually suffer from systematic dysfunction of multiple organs, but the most common and most severe phenotypes are neurodegenerative[1]. We assessed if overexpression of SNX8 in the brain can rescue disease phenotypes in *Hexb*[−/−] mice. For SNX8 overexpression, AAV particles carrying SNX8 were injected into the brain of P1 neonatal WT or *Hexb*[−/−] mice, with AAV-GFP as a control (Supplementary Fig. 5). AAV-infected mice were subjected to behavioral analyses at 3 months' age, and brain sections of these mice were subsequently examined for GM2 ganglioside (GM2, the primary storage substance of *Hexb*[−/−] cells) storage and NeuN[+] neuron density. *Hexb*[−/−] mice showed increased GM2 storage that peaked in the cerebellum (Fig. 5D) and loss of NeuN[+] neurons most severely found in the brainstem (Fig. 5E). Excitingly, SNX8 overexpression successfully rescued the storage of GM2 and the loss of NeuN[+] neurons in *Hexb*[−/−] mice (Fig. 5D, E). Moreover, behavioral assays revealed that *Hexb*[−/−] mice showed weakened muscles (Fig. 5F) and impaired body balance (Fig. 5G), assessed by grip test and rotarod. SNX8 overexpression fully rescued muscle weakness (Fig. 5F) and significantly restored body balance/muscle endurance (Fig. 5G). Altogether, these results strongly support the potential application of SNX8 as a therapeutic target for LSD treatments.

### Small molecule drugs enhance SNX8-lysosome interaction to suppress LSD phenotypes

As overexpression of SNX8 is currently impractical for clinical LSD treatment, we screened for potential SNX8-dependent drugs for LSD treatment using a natural compound library provided by TargetMol (Supplementary Fig. 6A). Three molecules, namely Elemicin, Isopsoralen and Morroniside (Supplementary Fig. 6B), were identified to reduce LSD phenotypes in all three LSD cell lines (Fig. 6A–E and Supplementary Fig. 6C–K). Importantly, these molecules failed to reverse LSD phenotypes in SNX8-KO cells (Fig. 6A–D and Supplementary Fig. 6C–G), suggesting that these molecules ameliorate the LSD phenotype in a SNX8-dependent manner. Further analyses showed that these three molecules enhanced the binding of SNX8 to lysosomes (Supplementary Fig. 7A, B) without influencing the expression level of SNX8 (Supplementary Fig. 7C). In agreement with this, Elemicin, one of the three molecules, was able to specifically increase binding of endogenous SNX8 to lysosomes when applied to cells (Supplementary Fig. 7D, E), and enhanced the tubulation of purified lysosomes in the in vitro tubulation assay in an SNX8-dependent manner (Supplementary Fig. 7F, G).

We next tested the effectiveness of the small molecules in mice. As three molecules behaved similarly, we chose Elemicin for experiments described below to assess if delivery of Elemicin to the brain can reverse disease phenotypes in *Hexb*[−/−] mice. As similarly found in human cells (Fig. 6A–E and Supplementary Fig. 6C–K), Elemicin profoundly suppressed LSD phenotypes in the *Hexb*[−/−] mouse fibroblasts (Fig. 6F, J). For Elemicin treatment in mice, 3 μl of 10 μM Elemicin in PBS was injected at a 2-day interval through a microinjection tube that was buried into the lateral ventricle of the brain of 2 months-old WT or *Hexb*[−/−] mice. GM2 ganglioside storage, NeuN[+] neuron density and behaviors of these mice were examined (Fig. 7A–E). Contrary to the peaked GM2 storage in the cerebellum (Fig. 7A, B) and loss of NeuN[+] neurons in the brainstem (Fig. 7C, D) in untreated mice, delivery of Elemicin successfully reduced the GM2 storage and restored the NeuN[+] neurons in *Hexb*[−/−] mice. Accordingly, Elemicin fully rescued muscle weakness (Fig. 7E) in *Hexb*[−/−] mice. These results suggest that Elemicin can serve as a potential anti-LSD drug.

### Discussion

Lysosomes are dynamic organelles and their functions rely heavily on trafficking and turnover. Majority of LSD phenotypes are actually related to secondary storage (e.g. cholesterol) and trafficking defects rather than the primary defect of the mutated gene. Indeed, TFEB overexpression was reported to suppress phenotypes of lysosome-related diseases through facilitating lysosome turnover[5,36–38]. In this report, we demonstrated that restoring lysosome reformation through overexpression of SNX8 or pharmacologically induced lysosomal recruitment of SNX8 could rescue/ameliorate LSD phenotypes (Fig. 7G) both in LSD model cell lines and LSD model mice (Figs. 5–7, Supplementary Figs. 4–7).

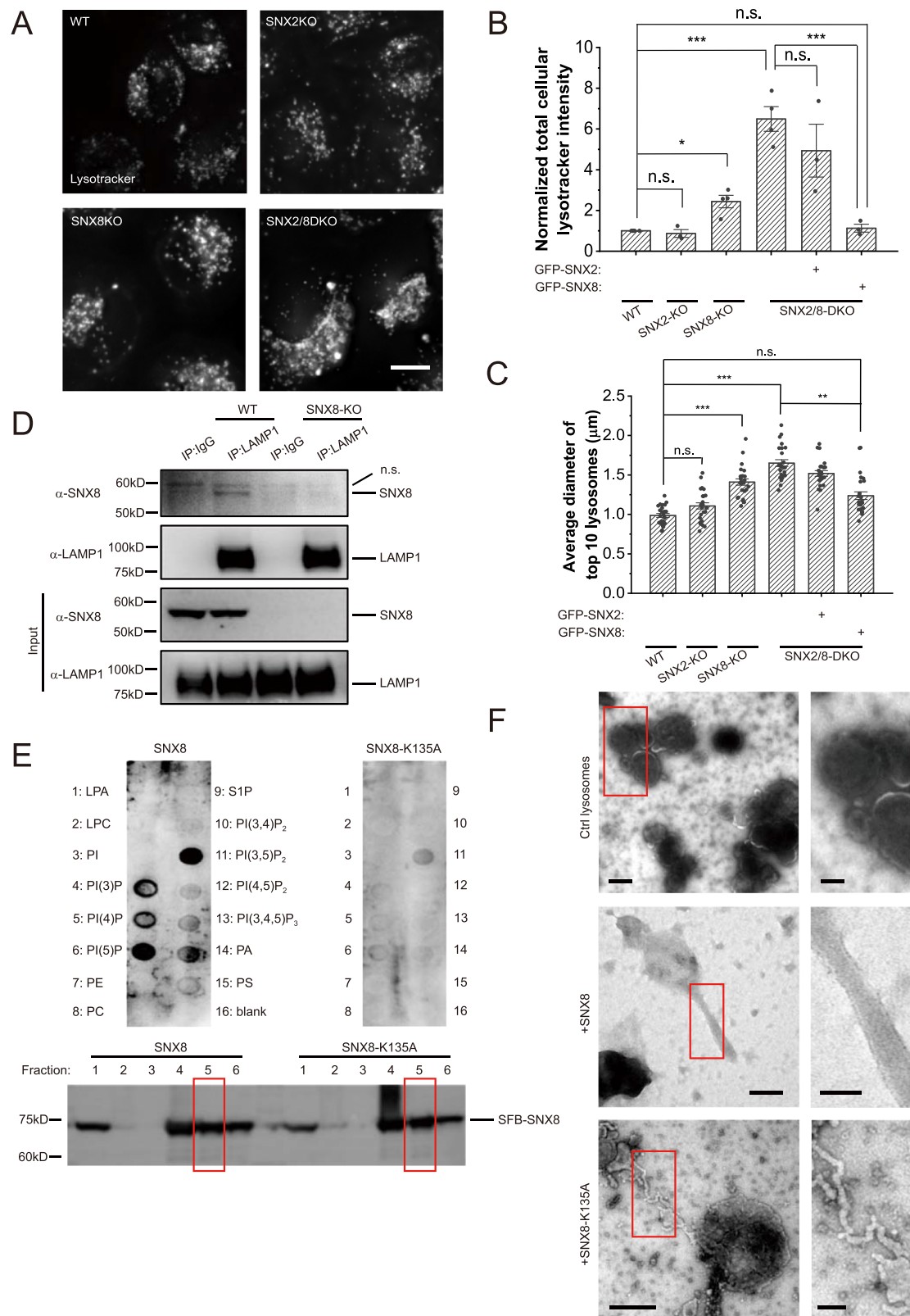

Our work is significant on two counts. First, we observed that the effect of SNX8 overexpression in rescuing LSD phenotypes in vitro or in vivo is almost complete in most examined aspects. Second, we identified three small molecules that targeted SNX8 to lysosomes that could mimic, to a certain extent, the overexpression of the SNX8 phenotype. As such, we found that Elemicin delivery to the brain of adult *Hexb*$^{-/-}$ mice produced satisfactory effects in

suppressing LSD phenotypes both at cellular and behavioral levels. Further chemical modifications of these molecules, which are beyond the scope of the current study, are needed to improve their efficacy. Furthermore, it is anticipated that small bioactive molecule drugs capable of enhancing SNX8 protein expression can replicate the positive effects observed with SNX8 overexpression. Additionally, it is speculated that SNX8-enhancing drugs could be used in

**Fig. 2 | SNX8 binds to and tubulates lysosomes. A** Representative images of WT, SNX2-KO, SNX8-KO, and SNX2/8-DKO HeLa cells stained with lysotracker for 30 min. Scale bar = 10 μm. **B, C** Quantifications of the total intracellular lysotracker intensity (**B**) and the size of largest lysosomes in cells (**C**) of cell groups shown in (**A**) ($n = 5, 3, 4, 4, 3, 3$, respectively, for each group for (**B**) and $n = 25$ for (**C**). For (**B**), $p = 1$ (WT vs. SNX2-KO), 0.027 (WT vs. SNX8-KO), <0.00001 (WT vs. SNX2/8-DKO), 0.28 (WT vs. SNX2/8-DKO + GFP-SNX8), 0.40 (SNX2/8-DKO vs. SNX2/8-DKO + GFP-SNX2), 0.00002 (SNX2/8-DKO vs. SNX2/8-DKO + GFP-SNX8). For (**C**), $p = 0.25$ (WT vs. SNX2-KO), <0.00001 (WT vs. SNX8-KO), 0 (WT vs. SNX2/8-DKO), 0.18 (WT vs. SNX2/8-DKO + GFP-SNX8), <0.00001 (SNX2/8-DKO vs. SNX2/8-DKO + GFP-SNX8)).

**D** WT and SNX8-KO HeLa cells were immunoprecipitated against LAMP1, and blotted against LAMP1 and SNX8. **E** SFB-SNX8 or SFB-SNX8-K135A were purified from HEK 293T cells and subjected to PIP Strips with indicated lipid dots. Fraction 5 (biotin-eluted) was used for the assay as indicated with red rectangles. **F** Surface scanning EM images of negative-stained lysosomes purified from SNX8-KO cells with or without 20 min incubation with 10 μM of SNX8 or SNX8-K135A proteins. Scale bars = 250 nm (left) or 100 nm (right). For graphs, error bars are s.e.m, statistical comparison was done using one-way ANOVA, Tukey test. Source data are provided as a Source Data file.

combination therapies alongside existing LSD treatments, such as enzyme replacement therapy (ERT), to achieve improved curative outcomes. These experimental findings strongly suggest that SNX8 represents a promising target for potential broad-spectrum LSD therapies. This has significant clinical implications, as the development of effective, broad-spectrum therapies for LSDs has been a long-standing goal due to the disease-specific nature of current therapeutic approaches. However, extensive testing across various LSD types will be necessary to fully assess the feasibility of SNX8 as a therapeutic target for LSDs.

Although both SNX2 and SNX8 may participate in lysosome tubulation, we have observed that unlike SNX8, SNX2-KO did not result in significant defects in tubulation or other LSD-related phenotypes, indicating that SNX2 is dispensable in the presence of SNX8 for lysosome tubulation under starvation conditions. This appears to be contradictory to a recent study by Rodgers et al. that reported the role of SNX2 in lysosome repopulation during basal autophagy[39], which investigated only SNX1 and SNX2 under nutrient-rich conditions. In the absence of tubulation-inducing conditions such as prolonged starvation or active phagocytosis, it is more likely that the reported study described normal lysosome fission events without the participation of lysosomal tubular structures, rather than induced lysosome tubulation. Nonetheless, further examinations across different cell types may be required to conclude the effect of SNX2 on lysosome tubulation/reformation, and our data on SNX2/8-DKO cells (Fig. 3) do indicate that SNX2 may serve as a functional redundant for SNX8 in lysosome tubulation.

Other than LSDs, lysosomal dysfunction or insufficiency in lysosomal clearance contributes crucially to the development of many neurodegenerative diseases (NDs)[40–42]. Boosting lysosomal functions can facilitate the clearance of harmful aggregations of disease-inducing proteins such as Aβ or α-synuclein. Indeed, recent studies showed that the enhancement of lysosome biogenesis through TFEB reduced the accumulation of aggregated proteins in ND models[13,43,44]. Interestingly, single nucleotide polymorphisms in the coding and intron regions of SNX8 were associated with late-onset Alzheimer's Disease (AD)[45]. In contrast, overexpression of SNX8 is associated with late-onset Alzheimer's Disease (AD)[46], although the mechanism underlying this effect was not well analyzed at the time. At the same time, SNX8 may reduce Aβ accumulation through rescuing lysosome reformation, which is worth further exploration. It is also conceivable that this rescue effect may apply to other NDs caused by the accumulation of protein aggregations, including the well-known Parkinson's Disease and Huntington's Disease.

Apart from lysosome-related functions, SNX8 has been reported to be involved in other cellular processes as well. Several reports indicate that SNX8 is involved in endosome-to-Golgi retrograde transport[27,47], while others suggest that SNX8 also participates in host defense against bacteria and viruses[48–50]. Another interesting study identified SNX8 as a potential inhibitor for de novo hepatic lipogenesis in NAFLD[51]. Thus, the availability of effective and safe SNX8-enhancing drugs will warrant further understanding on roles of SNX8 in other physiological processes and related diseases.

## Methods

This study complies with all relevant ethical regulations, no human research was conducted. Animal protocols used in the study were supervised by the Laboratory Animal Center of Zhejiang University (No. 19389).

### Molecular biology

cDNAs of SNX1, SNX2, SNX4, SNX5, SNX6, SNX8, SNX9, SNX32, and SNX33 were obtained from an on-site human cDNA library and transferred to pXF1EG plasmid and fully sequenced with EGFP-C primer. The K135A mutation of SNX8 was done using the following primer pair: F: GCCCTGCCACCCGCGGCAATGCTGGGAGCTG; R: CAGCTCCCAGCAT TGCCGCGGGTGGCAGGGC. The SNX8-BAR domain (aa 182-465) construct was made using the following primer pair: F: TTTTCTAG AATGGTGAGCAAGGGCGAGGAG; R: TTTGCTAGCCTAGTGAGGACAC AGGCCGTCCTC. The TGN38-mCherry construct was made by insertion of TGN38 cDNA into the pmCherryC1 plasmid, while LAMP1-mCherry was adopted from previous work[17]. To generate knockout cell lines, the CRISPR-Cas9 system was used, with the gRNA sequence transferred onto the PEPKO plasmid. Primers used for gRNA are as follows: SNX2-F: ACCGGCAGGAAGACTAGCTGGTTC; SNX2-R: AACG AACCAGCTAGTCTTCCTGCC; SNX8-F: ACCGCTGCGGCATCTGCA TTCGAC; SNX8-R: AACGTCGAATGCAGATGCCGCAGC; GLA-F: ACCG GCTCCCCAAAGAGATTCAGA; GLA-R: AACTCTGAATCTCTTTGGG-GAGCC; HEXA-F: ACCGTTTCCCCGCTTTCCTCACCG; HEXA-R: AAC CGGTGAGGAAAGCGGGGAAAC; NPC1-F: ACCGCTGGACACAGTAG CAGCAGG; NPC1-R: AACCCTGCTGCTACTGTGTCCAGC.

### Chemicals and Reagents

The Natural Compound Library used for the screen was purchased by the institutional facility from TargetMol (L6000, 2017 version), while Elemicin, Isopsoralen, and Morroniside were separately purchased from TargetMol. Filipin complex was purchased from XYbio. BODIPY™ FL C5-Lactosylceramide complexed to BSA and lysotracker were purchased from Thermo Fisher. PIP Strips were purchased from Echelon Biosciences. CytoTox 96 Non-Radioactive Cytotoxicity Assay kit for LDH release assay was purchased from Promega. Collagenase type II and Propidium Iodide (PI) were purchased from Sigma. Antibodies were purchased as follows: LAMP1 (CST 9091S, clone D2D11 and DSHB H4A3), SNX2 (BD, 611308, clone 13), SNX5 (abcam, ab5983, polyclonal), SNX8 (Life Span Biosciences, LS-C172487, clone 4F8), SNX9 (proteintech, 15721-1-AP, polyclonal), EEA1 (abcam, ab2900, polyclonal), GM130 (abcam, ab52649, polyclonal), RAB7 (abcam, ab137029, EPR7589), RAB5 (abcam, ab109534, EPR5438), Perilipin 2 (CST 95109 S, clone E6G6M), Complex II (abcam, ab109865, clone 4H12BG12AG2), GLA (Abcam, ab168341, clone EP5828(2)), HEXA (Proteintech, 11317-1-AP, polyclonal), NPC1 (Novus, NB400-148, polyclonal), NeuN (Abcam, ab177487, clone EPR12763), GM2 (Abcam, ab23942, polyclonal), GFP (Santa Cruz, sc9996, clone B2), FLAG (Sigma F3165, clone M2), Tubulin (Sigma T6074, clone GTU-88), GAPDH (Sigma G8795, clone GAPDH-71.1), Alexa-488-conjugated fluorescent antibodies were from Thermo Fisher. Routine chemicals and assay kits were purchased from Sigma and Thermo Fisher. FLAG, Tubulin, and GAPDH antibodies were used at 1:5,000, while all other primary antibodies were used at 1:1,000. For

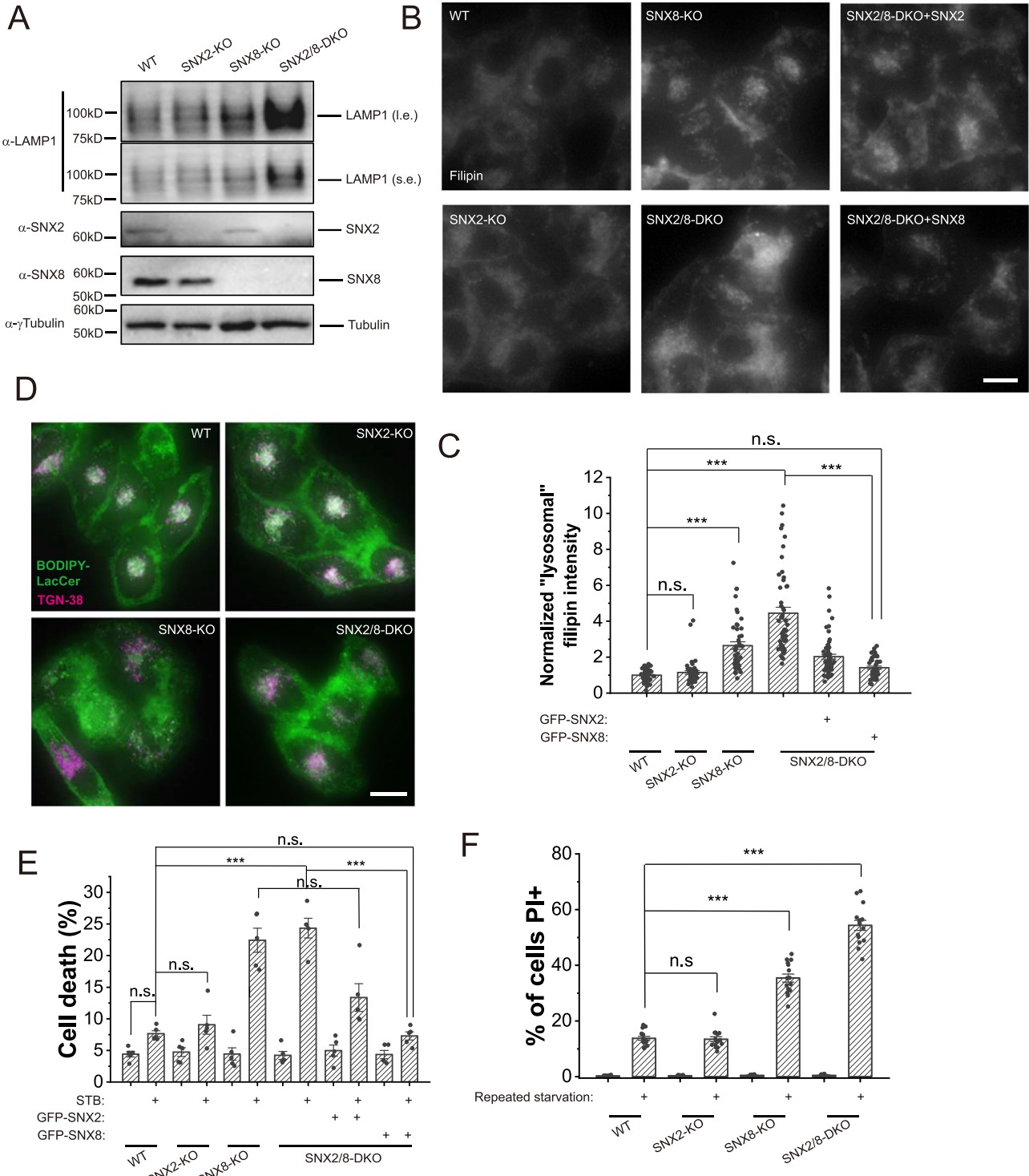

**Fig. 3 | SNX8-KO and SNX2/8-DKO cells show LSD-like phenotypes. A** Total LAMP1 levels in WT, SNX2-KO, SNX8 KO and SNX2/8-DKO HeLa cells were assessed with Western blots. **B** Representative images of HeLa cell lines as indicated fixed and stained with filipin to visualize free cholesterol in cells. Scale bar = 10 μm. **C** Quantification of intracellular filipin intensity (considered as "lysosomal" as it co-localized well with lysosomal marker LAMP1[17] for sample groups shown in (**B**) ($n$ = 53, 48, 46, 50, 61, 44, respectively, for each group, $p$ = 1 (WT vs. SNX2-KO), <0.00001 (WT vs. SNX8-KO), <0.00001 (WT vs. SNX2/8-DKO), 0.71 (WT vs. SNX2/8-DKO + GFP-SNX8), 0 (SNX2/8-DKO vs. SNX2/8-DKO + GFP-SNX8)). **D** Representative images of HeLa cell lines (as indicated) stably expressing TGN38 and loaded with BODIPY-LacCer, chased for 1 h. Scale bar = 10 μm. **E** HeLa cell lines (as indicated) were treated with STB buffer for 6 h to induce cell death, which was

assayed by LDH release assay ($n$ = 5 for all groups, $p$ = 0.72 (WT Ctrl vs. WT STB), 1 (WT STB vs. SNX2-KO STB), <0.00001 (WT STB vs. SNX8-KO STB), <0.00001 (WT STB vs. SNX2/8-DKO STB), 0.00098 (WT STB vs. SNX2/8-DKO + GFP-SNX2), 1 (WT STB vs. SNX2/8-DKO + GFP-SNX8), 1 (SNX8-KO STB vs. SNX2/8-DKO STB), 0.094 (SNX8-KO STB vs. SNX2/8-DKO + GFP-SNX2 STB), <0.00001 (SNX2/8-DKO STB vs. SNX2/8-DKO + GFP-SNX8 STB)). **F** HeLa cell lines (as indicated) were subject to a repeated serum starvation protocol with Glu-free DMEM (12 h starvation followed by 12 h complete medium, repeated 3 times). Cells were then stained with PI to visualize dead cells ($n$ = 15 fields for all groups, $p$ = 1 (WT stv vs. SNX2-KO stv), 0 (WT stv vs. SNX8-KO stv), 0 (WT stv vs. SNX2/8-DKO stv)). For graphs, error bars are s.e.m, statistical comparison was done using one-way ANOVA, Tukey test (two sided, no adjustments). Source data are provided as a Source Data file.

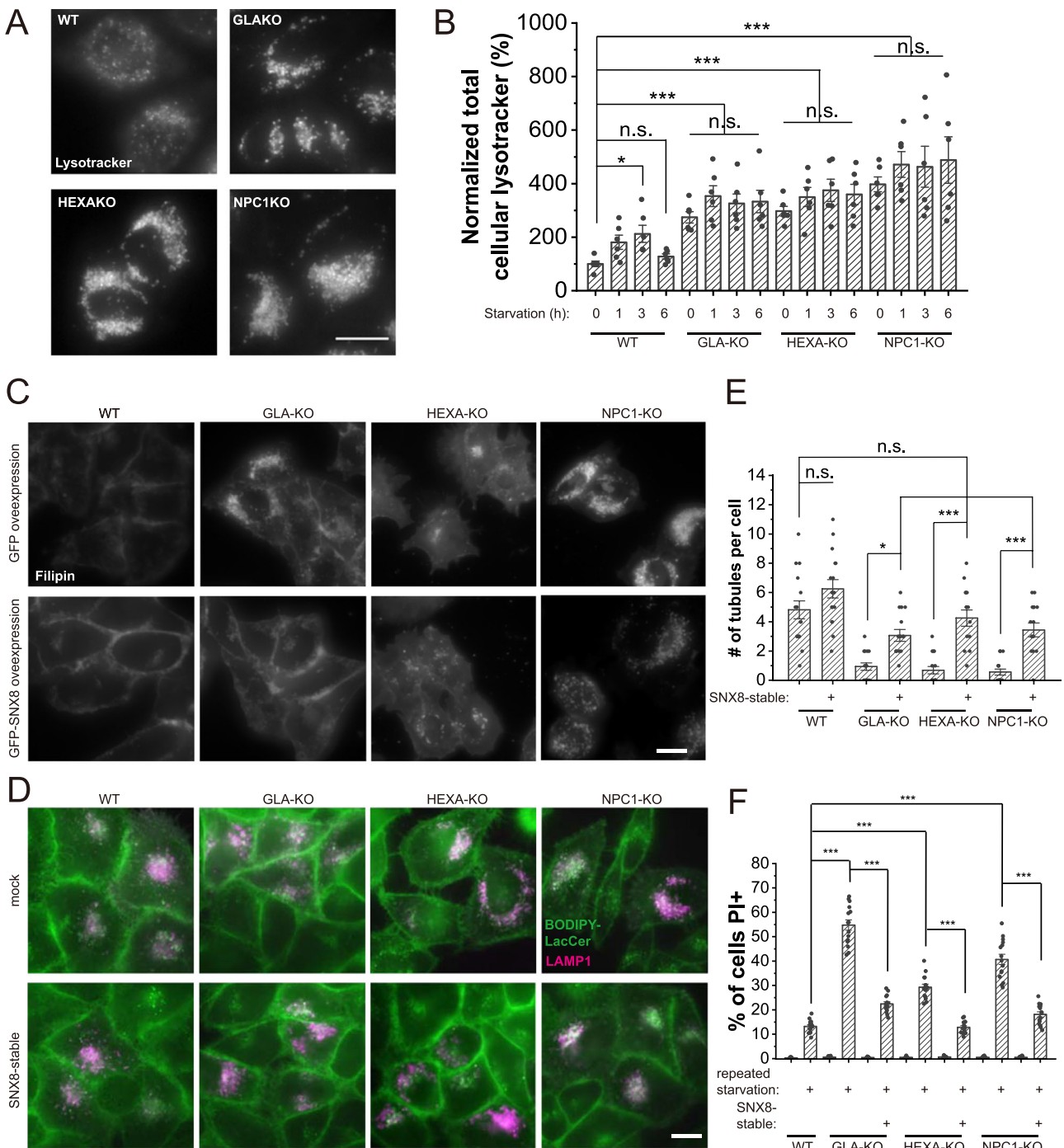

**Fig. 4 | SNX8 Overexpression rescues LSD phenotypes in LSD model cells.**
**A** Representative images of WT, GLA-KO, HEXA-KO, and NPC1-KO HeLa cells loaded with lysotracker to visualize lysosomes. Lysotracker positive puncta were enlarged in three KO cell lines. Scale bar = 10 μm. **B** HeLa cell lines (as labeled) were left untreated or treated with serum starvation for the indicated time and stained with lysotracker. The total lysotracker represents total lysosome volumes in cells ($n = 6$ for all groups, $p = 0.044$ (WT 0 h vs. WT 3 h), 1 (WT 0 h vs. WT 6 h), <0.00001 (WT 0 h vs. all LSD groups), 0.50 (GLA-KO 0 h vs. GLA-KO 1 h), >0.9 (GLA-KO 0 h vs. GLA-KO 3 h&6 h), 0.96 (HEXA-KO 0 h vs. HEXA-KO 1 h), 0.52 (HEXA-KO 0 h vs. HEXA-KO 3 h), 0.85 (HEXA-KO 0 h vs. HEXA-KO 6 h), 0.62 (NPC1-KO 0 h vs. NPC1-KO 1 h), 0.80 (NPC1-KO 0 h vs. NPC1-KO 3 h), 0.26 (NPC1-KO 0 h vs. NPC1-KO 6 h)). **C** WT, GLA-KO, HEXA-KO, and NPC1-KO HeLa cells stably expressing GFP or FLAG-SNX8 were fixed and stained with filipin to visualize free cholesterol. Blue channel (filipin) was shown in the images. Scale bar = 10 μm. **D** WT, GLA-KO, HEXA-KO, and NPC1-KO HeLa cells with or without stably-expressed SNX8 were transfected with LAMP1-

mCherry for 24 h, then loaded with BODIPY-LacCer, chased for 1 h and imaged. Scale bar = 10 μm. **E** WT, GLA-KO, HEXA-KO, and NPC1-KO HeLa cells with stably-expressed SNX8 were transfected with LAMP1-mCherry for 24 h and then serum-starved for 16 h before being subjected to a 1 min live imaging. The number of tubules within the 1 min period was quantified ($n = 16$ for all groups, $p = 0.33$ (WT ctrl vs. WT + SNX8), 0.12 (WT ctrl vs. GLA-KO + SNX8), 0.99 (WT ctrl vs. HEXA-KO + SNX8), 0.39 (WT ctrl vs. NPC1-KO + SNX8), 0.026 (GLA-KO ctrl vs. GLA-KO + SNX8), <0.00001 (HEXA-KO ctrl vs. HEXA-KO + SX8, 0.00043 (NPC1-KO ctrl vs. NPNC1-KO + SNX8)). **F** WT, GLA-KO, HEXA-KO, and NPC1-KO HeLa cells with stably-expressed SNX8 were treated with a repeated starvation protocol as in (Fig. 3F). Cells were then stained with PI to visualize dead cells ($n = 15$ fields for all groups, $p = 0$ for all comparison pairs shown). For graphs, error bars are s.e.m, statistical comparison was done using one-way ANOVA, Tukey test (two sided, no adjustments). Source data are provided as a Source Data file.

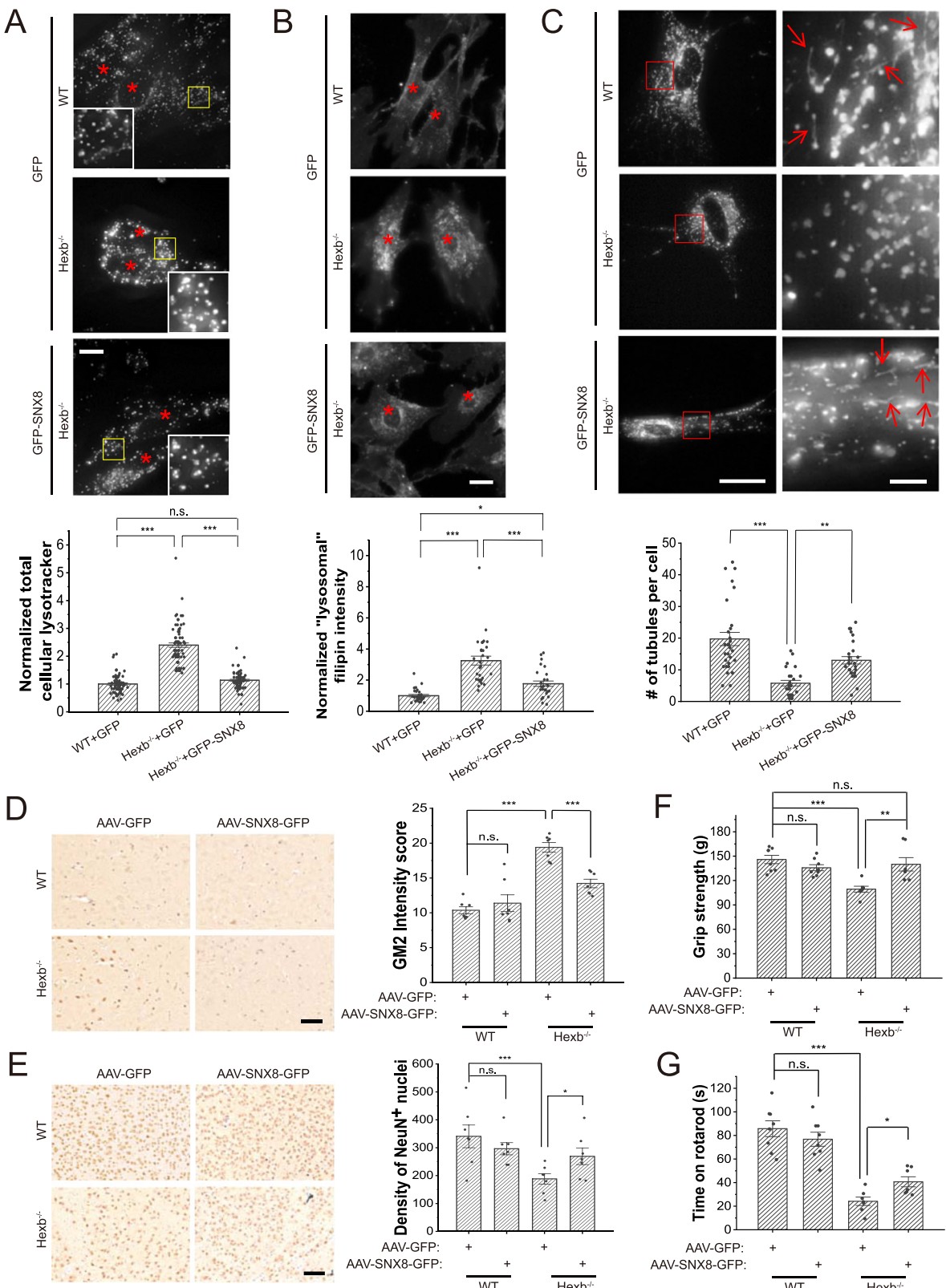

immunofluorescence, primary antibodies were used at 1:500. Secondary antibodies were used at 1:5,000 for Western blots and 1:1,000 for immunofluorescence.

## Cell Culture

Mammalian cells were cultured in a 37 °C incubator with 5% $CO_2$. HeLa (CCL-2) and HEK293T (CRL3216) cells were originally acquired from ATCC and were tested for mycoplasma contamination. HEK293T cells are on the list of frequently misidentified or cross-contaminated cell lines, and were used only for collection of expressed proteins, utilizing their fast growth curve. We did not authenticate this cell line as it was not used directly for any cell biology experiments. All other cell types used in the study are not listed as misidentified or cross-contaminated cells. HEK293T cells

**Fig. 5 | SNX8 overexpression rescues LSD phenotypes in _Hexb_$^{-/-}$ primary mouse fibroblasts and animals. A** WT or _Hexb_$^{-/-}$ primary fibroblasts isolated from P1 neonatal mice were transfected with GFP or GFP-SNX8 as indicated for 24 h and then stained with lysotracker. Representative images (upper) with red asterisks indicating cell nuclei and quantifications (lower) are shown ($n = 75$ for all groups, $p = 0.19$ for WT + GFP vs. Hexb$^{-/-}$+GFP-SNX8, and =0 for other two comparisons). Scale bar = 10 μm. **B** Sample groups same as in (**A**) were stained with filipin to visualize free cholesterol. Representative images (upper) with red asterisks indicating cell nuclei and quantifications (lower) are shown ($n = 30, 30, 28$, respectively, for each group, $p = 0.022$ for WT + GFP vs. Hexb$^{-/-}$+GFP-SNX8, and <0.00001 for other two comparisons). Scale bar = 10 μm. **C** Sample groups same as in (**A**) were transfected with LAMP1-mCherry then imaged for LAMP1. LAMP1-positive tubules were counted. Representative images (upper) showing the whole cell (left panels) and enlarged insets (right panels, ROI are marked with red boxes) and quantifications (lower) are shown ($n = 29, 25, 26$, respectively, for each group, $p < 0.00001$ for WT + GFP vs. Hexb$^{-/-}$+GFP-SNX8, and =0.0049 for Hexb$^{-/-}$+GFP vs. Hexb$^{-/-}$+GFP-SNX8). The fluorescent intensity was differentially boosted to clarify tubules' visualization

(pointed with red arrows). Scale bar = 20 μm for left panels and 3 μm for insets. **D**, **E** Representative images (left) and quantifications (right) of immunohistochemistry of GM2 ganglioside in the cerebellum regions (**D**), or NeuN in the brainstem regions (**E**) of 3-months-old WT and _Hexb_$^{-/-}$ mice injected with AAV-GFP or AAV-SNX8 ($n = 7$ for all groups, for (**D**), $p = 0.93$ (WT + GFP vs. WT + SNX8-GFP), and <0.00001 for the other two comparisons. For (**E**), $p = 0.55$ (WT + GFP vs. WT + SNX8-GFP), 0.00008 (WT + GFP vs. Hexb$^{-/-}$+GFP), 0.011 (Hexb$^{-/-}$+GFP vs. Hexb$^{-/-}$+SNX8-GFP)). Scale bar = 50 μm for (**D**) and 100 μm for (**E**). **F** Quantifications of the grip strength of forelimbs of 3-months-old WT and _Hexb_$^{-/-}$ mice injected with AAV-GFP or AAV-SNX8 ($n = 7$ for all groups, $p = 0.53$ (WT + GFP vs. WT + SNX8-GFP), 0.00033 (WT + GFP vs. Hexb$^{-/-}$+GFP), 0.87 (WT + GFP vs. Hexb$^{-/+}$SNX8-GFP), 0.0034 (Hexb$^{-/-}$+GFP vs. Hexb$^{-/-}$+SNX8-GFP)). **G** Same mice as in (**F**) were put onto rotating rotarods, and the duration they can stay on rotarods was recorded and quantified ($n = 7$ for all groups, $p = 0.31$ (WT + GFP vs. WT + SNX8-GFP), <0.00001 (WT + GFP vs. Hexb$^{-/-}$+GFP), 0.018 (Hexb$^{-/-}$+GFP vs. Hexb$^{-/-}$+SNX8-GFP)). For graphs, error bars are s.e.m, statistical comparison was done using one-way ANOVA, Tukey test (two sided, no adjustments). Source data are provided as a Source Data file.

were cultured in RPMI1640 medium + 10% FBS, while other cells were cultured in DMEM high glucose medium + 10% FBS. For fibroblast culture, neonatal mouse fibroblasts were isolated following a previously established protocol[17], and media were supplemented with Penicillin-Streptomycin (Thermo Fisher). Following the manufacturer's protocol, the transfections were done using Lipofectamine 3000 (Thermo Fisher). Cells were cultured in a normal medium to 60–80% confluency for STB treatment. Cells were then washed twice with STB (1% BSA, 140 mM NaCl, 1 mM CaCl$_2$, 1 mM MgCl$_2$, 5 mM Glucose, 20 mM HEPES, pH 7.4), then incubated with STB (100 μL for 96 well plates, 500 μL for 24 well plates).

### SFB-tagged protein purification
HEK293T cells expressing SFB-tagged protein were lysed (500 μL of lysis buffer per 10 cm dish). Lysates were centrifuged to remove pellets, and supernatants were mixed with Streptavidin sepharose beads (100 μL per 10 cm dish) and incubated at 4 °C for 4 h. Sepharose beads were pelleted by centrifuge (500 g, 2 min), and were washed three times with 500 μL lysis buffer. The supernatant was removed, and elution buffer (lysis buffer containing 1 mg/mL biotin) was added (150 μL per 10 cm dish) and incubated with agitation at 4 °C for 30 min. The supernatant was collected, and beads were resuspended with another 150 μL of elution buffer. Elutes from two elutions were combined. This elute was used for PIP Strip assays. For mass-spec, elutes were further affinity-purified with S-protein sepharose (40 μL per 10 cm dish, 4 °C for overnight). The supernatant was discarded through centrifugation at 500 g (5 min), and S-protein beads containing SFB-tagged proteins were mixed with Western blot loading buffer, boiled and applied for PAGE.

### Live imaging
Live imaging was carried out using a DV ELITE live imaging station (GE Healthcare) with a 37 °C heating closet and supply of CO$_2$. All still images were photographed with 0.8 μm step size z-stacking from the top to the bottom of cells. GFP, BODIPY-LacCer, and Alexa Fluor 488 were imaged using a standard FITC filter set, while mCherry, lysotracker and PI were imaged using a standard TRITC filter set. Filipin and DAPI were imaged using the DAPI filter set. For quantification of tubulation, each cell was imaged using a 1 μm X 5 z-stacking for 1 min at 2 sec per frame, and lysosomal tubules longer than 1 μm were counted over the 1 min period. For lysotracker staining, lysotracker stock was added at 1:10,000 (100 nM) to the cell medium and incubated for 30 min before imaging. PI was applied to cells at 1 μg/mL for 30 min before imaging for PI stain. Total cell number and PI positive cell number in images were counted using the cell counter tool in ImageJ by two separate researchers blind to the sample group information and averaged.

### Immunofluorescence and Western blot
Immunofluorescence and Western blot analyses were done following standard protocols. For immunofluorescence, cells were fixed with 4% PFA for 30 min and washed with TBST, then blocked in 2% BSA in TBST for 2 h, then incubated with primary antibody overnight at 4 °C, washed and incubated with secondary antibody for 2 h at RT. For Western blots, sample loading volumes were first properly adjusted (e.g. by housekeeping proteins). Samples were then run on a PAGE gel and transferred to PVDF membranes. The membranes were then blocked with either 2% BSA in TBST or 5% non-fat milk in TBST for 2 h before the application of desired primary antibodies for 2 h at RT. Blots were then washed and applied with corresponding secondary antibodies with HRP for 1 h, and visualized with Chemiluminescence. Primary antibodies were diluted to recommended ratios (generally, 1:500 for immunofluorescence, 1:5000 for Western blots with FLAG, tubulin, GAPDH, and actin, and 1:1000 for Western blots with other proteins).

### BODIPY-LacCer staining
BODIPY-LacCer staining was done using the following protocol: BODIPY-LacCer working solution was prepared with 5 μM BODIPY-LacCer-BSA in DMEM + 1% FBS and chilled on ice. Cells were washed with DMEM + 1% FBS, and the working solution was applied to the cells. Wait for 5 min to allow the solution to gradually warm up to RT, then transfer the plate to 37 °C and incubate another 40 min. Cells were then washed with DMEM + 1% FBS three times and incubated in the same medium for another 1 h before imaging. For quantification of BODIPY-LacCer staining results, number of dispersive puncta BODIPY signal, which is scattered BODIPY dots more than 2 μm away from the main perinuclear cluster, was counted for each cell. If no cluster is identifiable in the cell, then all BODIPY dots were counted.

### Filipin staining
Filipin staining was done using the following protocol: cultured cells were washed with PBS three times and then fixed with 3% PFA in PBS for 20 min at RT. Cells were then washed with PBS three times, and residual PFA was quenched with 1.5 mg/mL glycine in PBS for 10 min. Freshly-made filipin working solution (50 μg/mL filipin in PBS + 1% FBS) was then applied to cells for 2 h strictly in the dark. Cells were then washed with PBS 3 times in the dark and imaged using the DAPI filter set. Quantification of intracellular "lysosomal" filipin intensity was adopted from a previously described protocol[17]. Briefly, ImageJ was used to analyze the staining results, and for each individual cell, the fluorescence intensities of three randomly selected areas close to the cell border were averaged and considered intracellular background average intensity. Then, areas containing filipin puncta were selected avoiding the inclusion of non-puncta fluorescence (e.g. cloudy ER signals), and subtracted with intracellular background average to give the final value. Note that

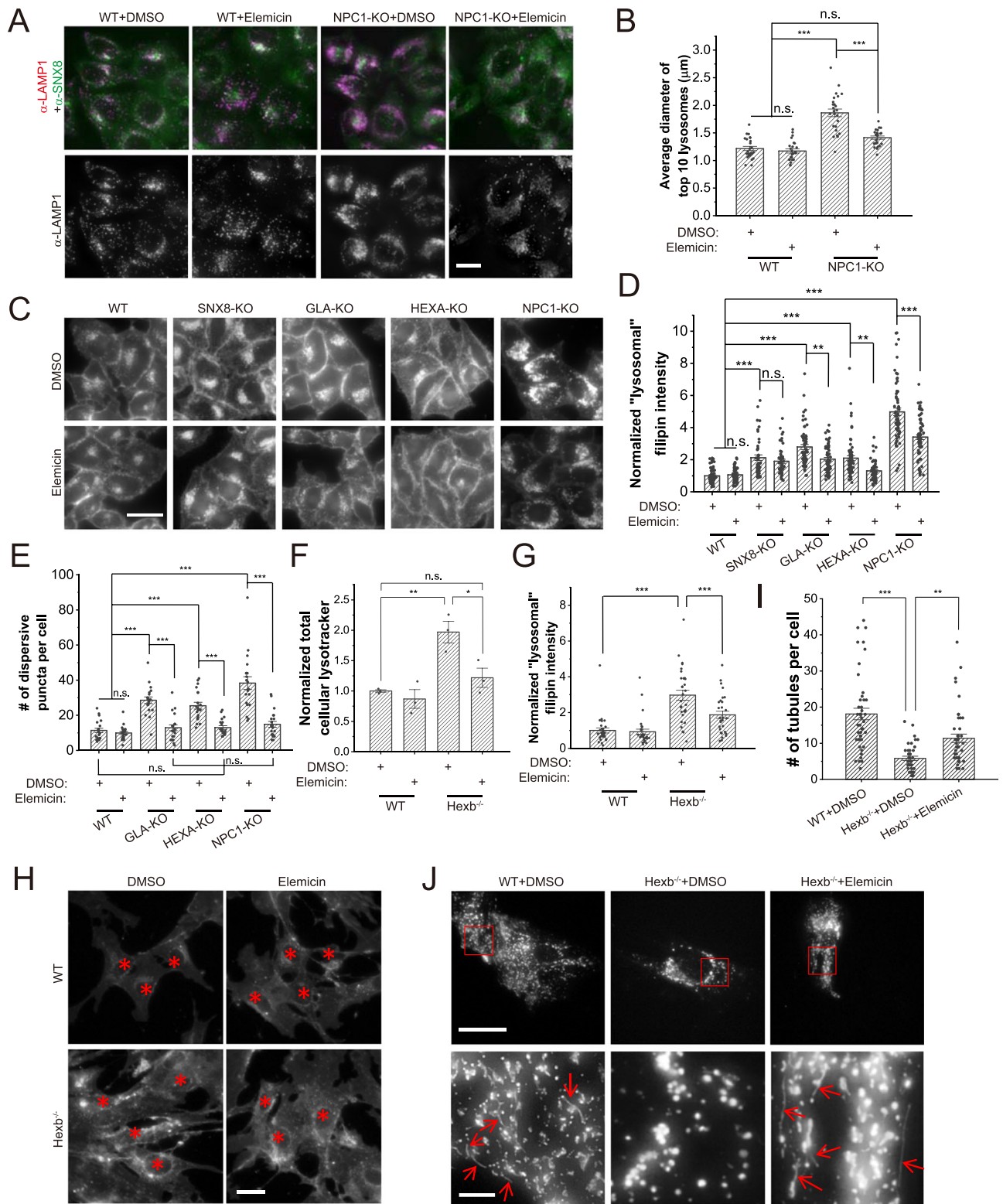

because lysosomes are the primary storage sites of un-esterified cholesterol under cholesterol storage conditions, the puncta filipin signal was presumed to be originated from lysosomal cholesterol[52].

## Cytotoxicity assay

Cell death was assayed using the CytoTox 96 Non-Radioactive Cytotoxicity Assay kit from Promega, following the kit's protocol. In short, cells were treated with indicated treatments on 96 well plates at 100 µL

per well with 3X well duplication. For each type of medium, one additional sample was treated as a control for the total cell amount. After treatments, 50 µL medium was collected from each well, and total control wells were added with lysis buffer at 1:10 30 min before collection. Reaction buffer and stop buffer (50 µL each) were then added to each well sequentially with 20 min reaction time, and light absorption at 490 nm was recorded. Cell death was then calculated as $(Abs_{sample} - Abs_{blank})/(Abs_{total} - Abs_{blank})$.

**Fig. 6 | Elemicin suppresses LSD phenotypes in LSD model cells and *Hexb*⁻/⁻ mouse fibroblasts. A**, **B** WT and NPC1-KO HeLa cells were treated with DMSO or Elemicin (10 μM) for 24 h, and then immunostained with anti-LAMP1 and anti-SNX8 antibodies. Diameters of large lysosomes visualized by LAMP1 were measured and compared. Representative images are shown in (**A**), and quantifications are shown in (**B**) (*n* = 25 for all groups, *p* = 0.89 (WT + DMSO vs. WT+Elemicin), 0.15 (WT + DMSO vs. NPC1-KO+Elemicin), and 0 for the other two comparisons). Note that in NPC1-KO cells, Elemicin also appeared to enhance the recruitment of SNX8 onto lysosomes. Scale bar = 20 μm. **C**, **D** Sample groups same as in (**A**) and (**B**) were stained with filipin to visualize free cholesterol. Representative images are shown in (**C**) and quantifications are shown in (**D**) (*n* = 58 for SNX8-KO cells and 64 for all other groups, *p* > 0.9 for WT + DMSO vs. WT+Elemicin and SNX8-KO + DMSO vs. SNX8-KO+Elemicin, <0.00001 for WT + DMSO vs. SNX8-KO + DMSO/GLA-KO + DMSO/HEXA-KO + DMSO/NPC1-KO + DMSO, 0.0044 (GLA-KO + DMSO vs. GLA-KO+Elemicin), 0.0021 (HEXA-KO + DMSO vs. HEXA-KO+Elemicin), <0.00001 (NPC1-KO + DMSO vs. NPC1-KO+Elemicin)). Scale bar = 20 μm. **E** WT, GLA-KO, HEXA-KO, and NPC1-KO HeLa cells were treated with DMSO or Elemicin (20 μM) for 24 h, then loaded with BODIPY-LacCer and chased for 1 h before imaging. Cells were then quantified with the number of dispersive BODIPY puncta. (*n* = 20 for all groups, *p* > 0.9 for all comparisons marked with n.s., =0.00002 for WT + DMSO vs. HEXA-KO+Elemicin, =0.00025 for HEXA-KO + DMSO vs. HEXA-KO+Elemicin, and

<0.00001 for all other comparisons). **F** WT or *Hexb*⁻/⁻ primary fibroblasts isolated from P1 neonatal mice were treated with DMSO or Elemicin (20 μM) for 24 h, stained with lysotracker, and total cellular lysotracker fluorescence was measured by flow cytometry (*n* = 3 for all groups, *p* = 0.0056 (WT + DMSO vs. Hexb⁻/⁻+DMSO), 0.71 (WT + DMSO vs. Hexb⁻/⁻+Elemicin), 0.023 (Hexb⁻/⁻+DMSO vs. Hexb⁻/⁻+Elemicin)). **G**, **H** Sample groups same as in (**F**) were stained with filipin to visualize free cholesterol. Representative images with red asterisks indicating cell nuclei are shown in (**H**) and quantifications are shown in (**G**) (*n* = 30 for all groups, *p* = 0 for WT + DMSO vs. Hexb⁻/⁻+DMSO, and =0.00085 for Hexb⁻/⁻+DMSO vs. Hexb⁻/⁻+Elemicin). Scale bar = 10 μm. **I**, **J** WT or *Hexb*⁻/⁻ primary fibroblasts isolated from P1 neonatal mice were transfected with LAMP1-mCherry and treated with DMSO or Elemicin (20 μM) for 24 h, serum starved for 16 h, and then imaged for LAMP1. LAMP1-positive tubules were counted (*n* = 45 for all groups, *p* = 0 for WT + DMSO vs. Hexb⁻/⁻+DMSO, and =0.0056 for Hexb⁻/⁻+DMSO vs. Hexb⁻/⁻+Elemicin). Representative images showing the whole cell (upper panels) and enlarged insets (lower panels, ROI are marked with red boxes) are shown in (**J**). Note that the fluorescent intensity was differentially boosted to clarify tubules' visualization (pointed with red arrows). Scale bar = 20 μm for upper panels and 3 μm for insets. For graphs, error bars are s.e.m, statistical comparison was done using one-way ANOVA, Tukey test (two sided, no adjustments). Source data are provided as a Source Data file.

## PIP Strip assay
PIP Strips were blocked with 3% BSA in PBST for 1 h at RT, and then incubated with 0.2 μg/mL affinity-purified SFB-tagged SNX2/8 proteins in blocking buffer for 1 h at RT. Strips were washed 3 times with blocking buffer and blotted with anti-FLAG antibody at 1:3000 in blocking buffer for 2 h at RT. Strips were then washed 3 times with PBST and detected with 45 min incubation of anti-mouse HRP-conjugated antibodies (Thermo Fisher).

## Small molecule library screening
Natural Compound Library (L6000, 2017 ver., TargetMol) was used for the screen. Each drug was applied to WT, SNX8-KO, and NPC1-KO HeLa cells at 10 μM for 48 h (medium was refreshed at 24 h). For WT cells, cell morphology was assessed visually, and cells were subjected to PI stain to determine viability. Drugs evoking elevated PI stain or the occurrence of noticeable morphological changes (usually cell shrinkage) are deemed toxic to cells. For SNX8-KO cells and NPC1-KO cells, cells were first stained with lysotracker and examined by flow-cytometry. Drugs that reduce lysotracker staining in NPC1-KO cells but not in SNX8-KO cells will then be tested for filipin staining. Drugs that reduce both lysotracker staining and filipin staining in NPC1-KO cells but not in SNX8-KO cells will be reconfirmed with GLA-KO and HEXA-KO cells. In summary, drugs that show no toxicity to WT HeLa cells, and significantly reduce lysotracker staining and filipin staining in GLA-KO, HEXA-KO and NPC1-KO, but not in SNX8-KO cells were selected as positive hits. Note: the EC$_{50}$ of the three molecules used in the manuscript in binding with SNX8 or alleviating LSD phenotype was not determined.

## Lysosome isolation
Lysosome isolation was done by adopting a previously established protocol with minor modifications in the last steps. Briefly, lysosomes were isolated through cell homogenization and two-step density gradient ultracentrifugation as described previously[53]. After obtaining the lysosome-containing fraction, a 150 mM NaCl, 20 mM HEPES, pH 7.4 buffer was added at a 1:1 ratio, and lysosomes were pelleted at 16,000 g for 5 min, then washed with the same solution. The pellet size was estimated visually by comparing with water drops of known volume in another tube, and lysosomes were reconstituted in 150 mM NaCl, 20 mM HEPES, pH 7.4 at roughly 1 mg/mL for in vitro binding assay with purified SNX8 proteins.

## Electron microscopy
For negative staining transmission electron microscopy to detect in vitro lysosome tubule formation, purified lysosomes (1 mg/mL) were

mixed with 10 μM purified SNX8 proteins and incubated at RT for 20 min. 3.5 μl of the sample was added to the grid coated with carbon for 30 sec, and the liquid was removed by filter paper. 3.5 μl staining solution (2% uranyl acetate in water) were added to the grid quickly for 30 sec and removed by filter paper. Specimens were viewed with a TecnaiG2 Spirit (Thermo Fisher) transmission electron microscope.

## Mouse behavioral assays
WT (C57BL/6 J) and *Hexb*⁻/⁻ (generated by GemPharmatech) littermate mice were either injected with 1.5 ×10¹³ viral genome/kg AAV-GFP or AAV-SNX8 (produced by OBiO Technology (Shanghai) Corp.) in the lateral ventricles at Day 1 postnatal or subjected to administration of 3 μl Elemicin (10 μM in PBS) or DMSO (1:1000 in PBS) through a buried tube in the lateral ventricle at once per two days for 30 days, beginning at 2 months of age. Both groups of mice were assessed with behavioral assays at 3 months of age at a roughly 1:1 sex ratio. Sex was not selected as Hexb is not a sex-linked gene and LSD phenotypes are not sex-linked. For rotarod, mice were placed onto the rotating cylinder, whose speed increased from 4 rpm to 40 rpm in 5 min. The total time the animal stayed on the cylinder was recorded. Animals repeated the experiment 6 times with a 5 min interval for rest. The animal was placed onto a clean metal grid fence for grip strength. The experimenter then lifted the animal's tail and let the animal grab the grid with its forelimbs. The experimenter then gently pulled up the animal until it failed to hold the grid, and the max grip strength was recorded. The experiment was repeated 8 times for each animal. Animals were bred in an on-campus laboratory animal center, with a 12h-12h day-night cycle, 20 °C temperature and 50% humidity, in SPF standard, negative-pressure secured clean rooms.

## Mouse brain slice immunohistochemistry
WT and *Hexb*⁻/⁻ littermate mice that have completed behavioral assays as described above were euthanized with isoflurane. Blood in the brain was removed through cardiac perfusion with PBS, and brain parts were cut and paraffinized. For immunohistochemistry, paraffinized tissues were sliced at 10 μm thickness, and sections were dewaxed with xylene for 10 min, then rehydrated with 100%, 95%, 90%, 85%, and 80% alcohol for 2 min sequentially. Endogenous peroxidase activity was quenched by 3% hydrogen peroxidase (VWR) in TBS for 30 min. Sections were then rinsed with PBS and blocked in 5% normal serum in PBST before incubation in the appropriate primary antibody: polyclonal rabbit anti-NeuN (1:500) or polyclonal rabbit anti-GM2 (1:200) diluted in 5% normal serum in PBST overnight at 4 °C. Sections were then rinsed in PBS and incubated with biotinylated anti-rabbit (Jackson, 1:1000) antibody

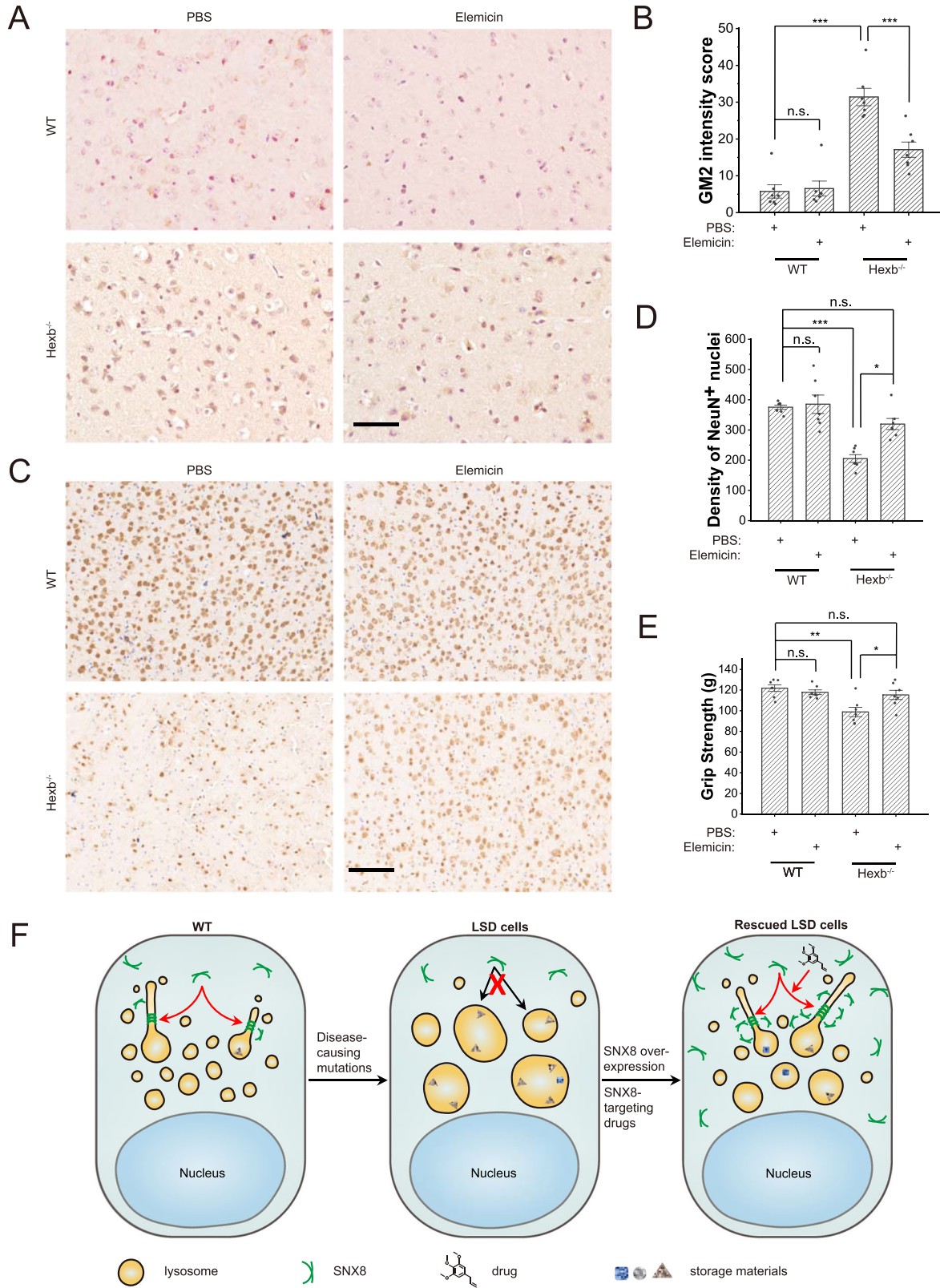

for 1 h at room temperature, washed with PBS, and followed by incubation in peroxidase streptavidin in PBS for 1 h. Then, sections were incubated in 0.05% DAB solution for 2–3 min to visualize immunoreactivity and counterstained with hematoxylin for one minute before dehydration in 95%, 95%, 100%, 100% alcohol for 5 min sequentially. Finally, sections were cleared in xylene and mounted with neutral resin. IHC images were analyzed with ImageJ.

## Statistics and reproducibility

Statistical data are presented as means ± standard error of the mean (s.e.m.). Origin 9.1 was used to process numerical data. P-values between experimental groups were calculated using analyses of variance (ANOVAs), and correlation coefficients were computed using respective functions in Origin. All data are generated from cells or mice pooled from at least three biologically independent experiments.

**Fig. 7 | Elemicin relieves neurological LSD phenotypes in *Hexb*$^{-/-}$ mice.**
**A, B** Representative images (**A**) and quantifications (**B**) ($n = 7$ for all groups, $p = 1$ for WT + PBS vs. WT+Elemicin, and <0.00001 for the other two comparisons) of immunohistochemistry of GM2 ganglioside in the cerebellum regions of 3-months-old WT and *Hexb*$^{-/-}$ mice treated with PBS or Elemicin for one month as described in the text. Scale bar = 50 µm. **C, D** Representative images (**C**) and quantifications (**D**) ($n = 7$ for all groups, $p = 0.83$ (WT + PBS vs. WT+Elemicin), 0.00040 (WT + PBS vs. Hexb$^{-/-}$+PBS), 0.39 (WT + PBS vs. Hexb$^{-/-}$+Elemicin, 0.037 (Hexb$^{-/-}$+PBS vs. Hexb$^{-/-}$+Elemicin)) of immunohistochemistry of NeuN in the brainstem regions of 3-months-old WT and *Hexb*$^{-/-}$ mice treated with PBS or Elemicin. Scale bar = 100 µm. **E** Quantifications of the grip strength of forelimbs of 3-months-old WT and *Hexb*$^{-/-}$ mice treated with PBS or

Elemicin ($n = 7$ for all groups, $p = 0.87$ (WT + PBS vs. WT+Elemicin), 0.0011 (WT + PBS vs. Hexb$^{-/-}$+PBS), 0.61 (WT + PBS vs. Hexb$^{-/-}$+Elemicin), 0.022 (Hexb$^{-/-}$+PBS vs. Hexb$^{-/-}$+Elemicin)). **F** A working model depicting how SNX8 rescues LSD phenotypes in disease cells. Dysfunctional lysosomal genes cause storage of lysosomal substances and secondary defects, accompanied by the loss of lysosome reformation. Up-regulation of SNX8 through overexpression or drugs (e.g. Elemicin) can restore lysosome reformation and facilitate storage substance clearance and lysosomal functions. For graphs, error bars are s.e.m, statistical comparison was done using one-way ANOVA, Tukey test (two sided, no adjustments). Source data are provided as a Source Data file.

Except for mouse behavioral assays, no data point was excluded. For behavioral assays, 6 individual runs on rotarods and 8 individual grips for grip strength tests were done for each mouse. The highest and lowest values for each assay were discarded before averaging.

## Reporting summary

Further information on research design is available in the Nature Portfolio Reporting Summary linked to this article.

## Data availability

All data points supporting the findings of this study are available within the paper and its Supplementary Information, source data are provided with this paper, including uncropped blots and gels. Original images of imaging data are also available on request, which can be addressed to Xin-Hua Feng (fenglab@zju.edu.cn).

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

## Acknowledgements

The authors thank Dr. Haoxing Xu for generously providing the LAMP1-mCherry plasmid. We are grateful to the Core Facility of Life Sciences Institute and its staff for assisting with various molecular and cellular analyses. We thank members of the Feng Laboratory for helpful discussion and technical assistance. This research was partly supported by grants rewarded to X-HF from MOST (2022YFC3401500), NSFC (U21A20356, 31730057, 91540205, 31571447) and ZNSF (LD21C070001), and the Fundamental Research Funds for the Central Universities. X.L. was supported by Postdoctoral International Exchange Program from China Postdoctoral Science Foundation and Zhejiang University, and Special Funds for Scientific Research Development of Zhejiang University (2021FZZX001-51).

## Author contributions

X.L. coordinated the study and drafted the manuscript. X.L. and C.X. performed most experiments and analyzed data. S.Z. generated LSD model cell lines and participated in analyzing small molecule screen results. J.G. was responsible for EM work. C.L., A.W., and J.C. helped with some animal experiments. Y.L. and D.N. offered essential reagents, experimental suggestions and critical reading of the manuscript. P.X. provided essential reagents and *Hexb*$^{-/-}$ mouse line. X-HF directed the project, supervised the overall study design, analyzed the data and wrote the paper.

## Competing interests

The authors declare no competing interests.
