## [Peer Review File · Nature Communications]

SNX8 Enables Lysosome Reformation and Reverses Lysosomal Storage DisorderREVIEWER COMMENTS

Reviewer #1 (Remarks to the Author):

This manuscript describes a role for SNX8 in promoting tubulation of late endocytic organelles. While this is not entirely novel (eg, PMID:34524084), it is built on in this paper as a novel approach to targeting lysosome storage disease (LSD) phenotypes. Overexpression or increased lysosomal recruitment of SNX8 partially rescues hallmarks of both cellular and animal models of LSD.

The manuscript is clearly written and nicely presents interesting findings but I do have some concerns. Broad concerns are around the therapeutic potential for SNX8, overexpression of which has previously been reported to result in NPC1-like phenotypes (PMID:24362679) and over-interpretation of effects on lysosomal tubulation. The resolution of the images shown is often insufficient to identify lysosomal tubules, which sometimes appear extremely long here (eg 12.5um, ie the size of approx 25 lysosomes). My feeling from the data shown is that though SNX8 likely does mediate endo/lysosomal tubulation, as has been previously described, the images presented here instead show networks of recycling vesicles, which in no way detracts from the importance but would be strengthened by more data on the mechanism of SNX8 expression/activation on cargo recycling and retrograde transport in LSD models - is retromer involved for example? Specific comments:

1. In Supp Fig1E, it should be made more clear which colour is SNX8 and LAMP1, rather than leaving the reader to assume that SNX8GFP is green and LAMP1mCh red. Also red should be changed for magenta to improve colour-vision accessibility. The "tubule" highlighted in SNX8 KO +SNX8 is very long, but mostly appears to be green with discreet LAMP-1 positive puncta, suggesting that rather than being a lysosome tubule, maybe it is a string of lysosome-derived recycling vesicles along a SNX8 +ve microtubule? Ideally you would do some correlative light and electron microscopy on these structures but that isn't always feasible.
2. Figure 2A, the resolution is not sufficient to comment on the size of individual lysosomes. There is clearly expansion of the lysosomal population, but from these images it is not clear if that is enlargement of individual lysosomes or increased numbers and clustering. Data shown in Supp Fig1E was much more clear and I would suggest should be swapped with Fig2A.
3. LysoTracker quantitation shown in Supp Fig2 should be in the main figure2 and representative images should be shown for Supp Fig2C.
4. Data supporting purity of "purified lysosomes" used in Figure 2C should be shown in Supp Fig2. Are they positive for other endosomal markers (ie, early v late endosome markers, eg Rab5/EEA1 v RAB7/LBPA)? What about other SNXs or known SNX binding proteins? Lipid droplet markers (see below)?

5. Fig 2E, the control lysosomes look more like lipid droplets and are there any lysosomes in the +SNX8 panel? Clathrin has been implicated in budding from lysosomes and there could be a hint of a clathrin lattice on the +SNX8 mutant lysosome?
6. Fig 3D shows a reduction in transport to the Golgi but it is impossible to say if lysosome to Golgi transport is disrupted without showing a lysosomal marker as well.
7. Fig 4D-F is not discussed in the text! 4D should have lysosomal and TGN markers co-stained and representative images/movies of tubule formation should be accompany 4E.
8. From the images shown in 5A, it can't be determined if lysosomes are enlarged. Increased lysotracker signal could be due to increased lysosome numbers/clustering or even reduced pH.
9. Supp Fig 6A shows some very nice blots consistent with increased SNX8-lysosome interaction following treatment with drugs, but a no lysosomes control should be included to show that the drugs aren't precipitating SNX8 in the pellet. It would be nice to see IPs of endogenous protein from whole cells and/or immunofluorescence of SNX8/lysotracker \pm the three molecules.
10. Supp Fig 6C, there appears to be a dose-dependent reduction in SNX8 with elemicin - it might be worth checking effects of treating with lower concentrations (eg 10uM).

Minor points:

- Figure 1, the rationale for the starve condition should be explained in the text.
- p7, line 21, I think should read SNX8 could bind PI(5)P and PI(3,5)P2.

Reviewer #2 (Remarks to the Author):

In this manuscript, the authors propose that the sorting nexin SNX8 is required for lysosome reformation, and that enhancing SNX8 function may be a potential therapeutic approach for treating lysosomal storage disorders. SNX8 knockout leads to lysosome swelling, reduced lysosome tubulation and cholesterol accumulation, which is restored by recombinant SNX8 expression. SNX8 overexpression, or small molecules that promote SNX8 function, reverse lysosome swelling in several LSD cellular models, and restore muscle function in a Hexb knockout mouse model of gangliosidosis. Overall, these findings are of significance to the cell biology field as lysosome reformation is an area of great interest. However, the physiological data is fairly limited, and it is challenging to understand how enhanced lysosome reformation would restore the defective lysosomal enzyme function of LSDs. Some aspects of the methodology for measuring lysosome reformation also need improvement, and it is not clear how SNX8 is recruited to lysosomes.

Major comments

1. Although LSDs caused by loss of lysosome enzyme function can exhibit reduced lysosome reformation, it is unclear how SNX8-dependent lysosome reformation would restore lysosome function and improve disease pathogenesis as newly formed lysosomes would also be expected to retain defective enzyme function. This is likely to limit the clinical applications of SNX8 as a therapeutic target. SNX8 activation appears to acutely alleviate GM2 accumulation and motor function in vivo, but it is difficult to determine from the data presented here whether disease is reversed as there is no evidence of impact on long term histopathology, motor function or survival of LSDs. Similarly, cell line rescue experiments are performed by transient transfection of GFP-SNX8, does stable GFP-SNX8 expression elicit the same sustained effect?

2. Much of the authors' data relies on changes in the intensity and/or size of lysotracker, a dye that labels acidic cellular compartments and is commonly used to estimate total lysosome volume. However, as defective lysosome reformation leads to not only enlarged lysosomes but also reduced lysosome numbers (PMID: 20526321, 28857423, 33119550), the authors also need to determine whether there is a corresponding change in lysosome numbers with SNX8 KO or overexpression, or with SNX8-activating small molecules. As lysotracker does not label terminal storage lysosomes (ie nascent lysosomes generated by reformation) (PMID: 28857423, 20526321, 27498570), this analysis should be performed with LAMP1 or LAMP2 antibody staining to label all lysosomes.

3. As the authors acknowledge, cells undergo lysosome reformation from several hybrid organelles (eg autolysosomes, endolysosomes) in response to different stimuli, and distinct molecular machinery facilitate reformation from these different compartments (PMID: 33950241). Which hybrid organelles does SNX8 regulate tubulation from, and is this required for the lysosomal degradation of particular endocytic and/or autophagic cargoes? Is this function nutrient dependent or independent?

4. The authors claim that SNX8 is recruited to lysosomes through its interaction with PI(5)P and/or PI(3,5)P₂, but they provide little evidence for this. The authors show that mutating the K135 residue reduces PI(3,5)P₂ and PI(5)P binding in vitro, but this mutant generates thinner, distorted tubules of purified lysosomes, suggesting this mutation may instead affect SNX8 membrane deformation function. What is the rationale for mutating this particular residue? The authors need to determine whether this mutation affects SNX8 localization to lysosomes, and whether interfering with PI(3,5)P₂ or PI(5)P levels affects wild-type SNX8 localization. The authors should also examine whether expression of the K135A mutant in SNX8 KO restores lysosome number and size.

5. The data showing that SNX8 loss enhances cell death with repeated serum starvation is not particularly convincing, as very little cell death seems to occur under these conditions (only 2% of cells are dead in starved SNX8 KO sample). The authors should instead perform starvation with glutamine-free

DMEM or EBSS/HBSS treatment, which may induce more widespread cell death (PMID: 26139536, 35968799).

6. There is insufficient evidence to conclude whether elemicin, isopsoralen or morroniside enhance SNX8 localization to lysosomes. In Figure 7A, there are no loading controls to show whether there is equal lysosome input. There is no statistical analysis of the graph in Figure 7B, and the legend does not state what the quantification was normalized to. Changes in SNX8 localization to lysosomes should also be confirmed by immunofluorescence experiments. The authors should also determine whether these drugs affect SNX8 function at endosomal compartments, as has been previously described (eg PMID 19782049, 34524084).

7. Some further *in vivo* characterization is required. Were the behavioural studies (grip strength and rotarod experiments) performed by investigators blinded to treatments? This is critical given the potential for experimenter bias when performing rodent behavioural-type experiments. Does recombinant SNX8 expression or drug treatment also restore the level of apoptosis, lysosome homeostasis or brain histopathology as previously reported for HexB KO mice (PMID: 9302266, 7550345, 8789434)? The authors would also need to show that recombinant SNX8 expression or drug treatment restores lysosome numbers *in vivo*. In control studies, GFP tissue staining should be performed to show the proportion of cells expressing GFP or GFP-SNX8, and confirm the specificity of GFP-SNX8 expression in brain versus other tissues.

Minor comments

1. Another group recently reported that SNX2 is recruited to endolysosomes by PI(3,5)P2 and promotes lysosome reformation (PMID: 35968799). The authors should acknowledge this study and speculate as to why they observe more modest effects with SNX2 KO in their cellular models.

2. In Figure 1D, separated LAMP1 and SNX2/8 channels should also be included as LAMP1-positive tubules are difficult to see in the merged images.

3. Figure S1E should be moved to main figures with the corresponding quantification (Figure 1E). A GFP-vector control should also be included, as plasmid transfection may effect lysosome homeostasis. GFP-SNX8 expression does not appear to restore the swollen lysosome size observed in SNX8 or SNX2/8 KO cells.

4. In Figure 2B, there are multiple SNX2 bands per lane and some are present in the IgG control.

5. Images of starved cells in Figure S2B should also be included.
6. In Figure S2C, statistical comparison between each cell line at the 6 hour timepoint needs to be performed to draw appropriate conclusions.
7. How did the authors quantify the lysosome-specific pool of filipin versus other cellular pools? Lysosome co-labeling should be shown for all the lysosomal filipin experiments (Figure 3B, 4C, 6C).
8. On page 19, line 18, should “50 L” be “50 μ L”?
9. For lysosome isolation experiments, how did the authors confirm there were no other contaminating organelles present?
10. All figures should show individual data points, and legends should state the number of cells or mice analyzed.
11. Many graphs have axis breaks where inappropriate (Figure 1B, 5G, 7E, S3B).
12. Immunofluorescence and immunoblotting protocols should be properly described in methods section.

Reviewer #3 (Remarks to the Author):

The manuscript by Xinran Li and colleagues reports that SNX8 – and to a lesser extent SNX2 – has a role in tubulation of lysosomal membrane when overexpressed. SNX8 is one of several PX-BAR family members proteins with known roles in tubulation of membranes in the endomembrane system. Loss of SNX8 (by knockout) leads to an enlarged lysosomal compartment and several trafficking and localization defects, consistent with those seen in cell models of lysosomal storage disorders. The authors point out one avenue for targeting lysosomal storage disorders may be improving lysosome reformation, which is known to require membrane tubulation. The authors show that overexpression of SNX8 reversed some of the storage-disorder-like phenotypes in cultured cells as well as in brains of a mouse model of a

lysosomal storage disorder. The authors propose that improving lysosome reformation by, for example, augmenting the function of SNX8 (shown in this work by overexpression of SNX8) may be an approach that can be used to target and treat several lysosomal storage disorders. The authors finish this work by identifying 3 natural products that enhance SNX8 association with lysosomal membranes and reversed storage disorder phenotypes in cultured cells and their mouse model. These products did not alter SNX8 expression.

Overall, the central hypothesis is attractive and straightforward, and the authors present some intriguing data suggesting that strategies to enhance SNX8 tubulation function may aid in targeting storage disorders. However, there are a number of shortcomings.

The authors show that overexpression in COS7 cells leads to some colocalization with the standard lysosomal marker LAMP1. The authors test only SNX-BAR family members, most of which have previously been described to be associated with the endosomal compartment. Clearly, overexpression may lead to significant changes in the steady state distribution of SNX2 and SNX8. The contrast with endogenous SNX2 and SNX8 is clear: the images shown in 1C show essentially no (SNX2) and perhaps some (at best) colocalization with SNX8. The authors may wish to conduct a more complete assessment and quantification of these proteins with EEA1 and certainly determine the correlation coefficients of endogenous SNX2 and SNX8 with LAMP1 to convey the changes brought on by overexpression. A more complete characterization of this type may be of interest or necessary in view of the recent study (Suzuki and colleagues, 2021) demonstrating that highly expressed SNX8 in HeLa cells resulted in tubulation from endosomes. In the authors' system, is the endosomal compartment still distinct on SNX8 overexpression?

The negative stain electron microscopy shown in Fig. 2E is uninterpretable in its current form. The authors use purified lysosomes and incubate it with purified SNX8 or a point mutant. It is not clear from the figure, legend, or associated text whether the lysosomes are purified from SNX2 or 8 knockout cells or not – is endogenous SNX8 or SNX2 present? The role of the purified SNX8 in the tubulation is hard to assess in this context without a demonstration (immunogold?) that it indeed associates with the purified lysosomes. Perhaps a model liposome system using PI(3,5)P2 shown to bind SNX8 in Figure 2D is one approach. Perhaps the authors could also show the quality of their SNX8 purification. Indeed, the result with the mutant K135A mutant, showing formation of thinner tubes than wild type, is unexpected. In other systems, mutations of phosphoinositide-binding residues in the PX domain blocks tubulation of model membranes entirely.

The data with elemicin, isopsoralen and morroniside showing enhancement of SNX8 recruitment to lysosomes is interesting and perhaps a little surprising as the structures of the drugs shown in Sup Fig 5B are quite different. All stimulate binding of (HeLa endogenous) SNX8 and show increased binding, albeit to differing extents. It would be interesting to see whether SNX2 or other SNX proteins that don't rescue lysosomal storage disorder phenotypes or that don't colocalize with LAMP1 when overexpressed have this effect.

Smaller issues:

There are some significant factual inaccuracies in this work:

Page 5 Line 12: “The majority of the SNXs carry a BAR domain capable of inducing membrane curvature”. The majority of SNX proteins are not SNX-BARs and not all SNX-BARs can generate membrane curvature. The Collins lab has pointed out that the human genome has 49 PX domain-containing proteins (most are also known as SNX proteins) (Chandra and others, 2019) but only 12 are PX-BARs, or SNX-BARs (van Weering and others, 2010).

Page 7 line 20: “SNX proteins are known to bind phosphoinositides through their PX domains”. This is not the case. The PX domains of SNX5/6 and SNX32 are cargo binding and don’t bind phosphoinositides (for example, Chandra and colleagues, 2019; Yong and colleagues, 2020)

The data presented in Fig. 1B shows that SNX1 (not pursued) and SNX2 show an enhanced correlation coefficient with LAMP1 on overexpression. It is curious that their obligate dimerization partners (SNX 5 or 6) show lower correlation coefficients with LAMP1 than even the GFP control. Could the authors comment on this? This may be less a concern with SNX8 as it homodimerizes, but it indicates that there may be some challenges associated with overexpressing SNX-BARs.

Page 9 line 12: “These cell lines all showed similar LSD phenotypes (Fig. 4 & Supplemental Fig. 4B, 4C)”. The lysosomal morphology of GLAKO, HEXAKO and NPC1KO cells are not similar though they do all look increased compared to WT cells.

Manuscript NCOMMS-22-45911

Point-by-point reply to the reviewers' comments:

We thank the Editor and Reviewers for their efforts in evaluating our manuscript, and for the keen and constructive comments. Through careful experiments in response to Reviewers' comments and concerns, we believe the revised manuscript has answered all the questions raised by the reviewers and feel that the manuscript is now more clearly delivered, and major conclusions are strengthened. We have made the following major changes in the revised paper (with figures, figure legends and text marked up with **yellow highlight** in the manuscript files):

Please note that in the revised manuscript the following are new revised figure panels with either completely new data or experiments being redone with more controls:

Fig. 1E, F

Fig. 2B, C, F

Fig. 3F

Fig. 4C-F

Fig. 5A

Fig. 6E

Fig. S1C

Fig. S2B, C, E

Fig. S3D

Fig. S4C, D, E

Fig. S6A, B, D

In addition, we have included **20 "Rebuttal Figures"** to explain or answer the concerns raised by the reviewers. Since we feel these data will not advance our conclusions (on top of space restraint), we have decided not to include them in the revised manuscript.

In the following pages, we have a point-to-point reply to the editor's summary and each reviewer's comments.

Reviewer #1

This manuscript describes a role for SNX8 in promoting tubulation of late endocytic organelles. While this is not entirely novel (eg, PMID:34524084), it is built on in this paper as a novel approach to targeting lysosome storage disease (LSD) phenotypes. Overexpression or increased lysosomal recruitment of SNX8 partially rescues hallmarks of both cellular and animal models of LSD.

The manuscript is clearly written and nicely presents interesting findings but I do have some concerns. Broad concerns are around the therapeutic potential for SNX8, overexpression of which has previously been reported to result in NPC1-like phenotypes (PMID:24362679) and over-interpretation of effects on lysosomal tubulation. The resolution of the images shown is often insufficient to identify lysosomal tubules, which sometimes appear extremely long here (eg 12.5um, ie the size of approx 25 lysosomes). My feeling from the data shown is that though SNX8 likely does mediate endo/lysosomal tubulation, as has been previously described, the images presented here instead show networks of recycling vesicles, which in no way detracts from the importance but would be strengthened by more data on the mechanism of SNX8 expression/activation on cargo recycling and retrograde transport in LSD models - is retromer involved for example? Specific comments:

Response: We appreciate the detailed comments from the Reviewer. Concerning the possible involvement of retrograde transport in the SNX8-LSD phenotype axis, we knocked out VPS35, a key component of the retromer complex, in WT or GLA-KO/HEXA-KO/NPC1-KO HeLa cells (**Rebuttal Fig. 1**), and analyzed LSD phenotypes in these cells compared to their VPS35-intact counterparts, and the effect of Elemicin or SNX8 expression in these cells (**Rebuttal Fig. 2-5**, next pages).

Rebuttal Fig. 1

Results showed that VPS35-KO cells showed apparent lysosomal membrane trafficking defects including enlarged lysosomes, increased cholesterol storage and impaired

retrograde transport of BODIPY-LacCer (Rebuttal Fig. 2-4 in the immediate following pages). However, degrees of defects in GLA/VPS35-DKO, HEXA/VPS35-DKO and NPC1/VPS35-DKO cells were somewhat additive, suggesting that VPS35-KO disrupts a parallel pathway from GLA-KO/HEXA-KO/NPC1-KO regarding lysosomal membrane trafficking defects, supported by the fact that treatment with Elemicin or SNX8 overexpression in these DKO cells did not rescue the VPS35-KO portion of the defect (Rebuttal Fig. 2-4). Moreover, VPS35-KO cells did not result in defects in lysosomal tubulation or viability under repeated/severe starvation (Rebuttal Fig. 5). Taken together, our data suggest that retromer participates in housekeeping lysosome membrane trafficking, but not in lysosome reformation or cell viability under nutrient depletion, both of which are regulated by SNX8. Also, our data suggest that the rescue of LSD phenotypes by SNX8 activity does not involve the retromer.

These rebuttal figure images are presented here in large sizes for easy visualization.

Rebuttal Fig. 2

Rebuttal Fig. 3

Rebuttal Fig. 4

Rebuttal Fig. 5

1. In Supp Fig1E, it should be made more clear which colour is SNX8 and LAMP1, rather than leaving the reader to assume that SNX8GFP is green and LAMP1mCh red. Also red should be changed for magenta to improve colour-vision accessibility.

Response: We thank the Reviewer for the suggestion and have made corresponding changes.

The "tubule" highlighted in SNX8 KO +SNX8 is very long, but mostly appears to be green with discreet LAMP-1 positive puncta, suggesting that rather than being a lysosome tubule, maybe it is a string of lysosome-derived recycling vesicles along a SNX8 +ve microtubule? Ideally you would do some correlative light and electron microscopy on these structures but that isn't always feasible.

Response: We agree with the Reviewer that these tubules appear mostly green in the merged images. It is worth noting, though, that LAMP1 signal on lysosomal tubules is low compared with the vesicular LAMP1 signal. In LAMP1-only images, exposure is boosted to clearly visualize LAMP1-positive tubules, but in multi-channel images, such image processing will result in unbalanced color-combination. As can be seen in **Rebuttal Fig. 6** (next page), when the red channel is separated and intensity is boosted, LAMP1 signal can be seen on these tubules. It is worth noting that we do not exclude that some of these long tubules may indeed be endosomal tubules. We have also tried to stain LAMP1 and observe these tubules under electron microscopy (EM), but the stain protocol seems to disrupt lysosomal tubules and we failed to observe immunogold-stained tubular structures under EM.

Rebuttal Fig. 6

2. Figure 2A, the resolution is not sufficient to comment on the size of individual lysosomes. There is clearly expansion of the lysosomal population, but from these images it is not clear if that is enlargement of individual lysosomes or increased numbers and clustering. Data shown in Supp Fig1E was much more clear and I would suggest should be swapped with Fig2A.

Response: We thank the Reviewer for the suggestion. However, Fig. S1E was the corresponding images of Fig. 1E, and we have thus moved Fig. S1E to the **new Fig. 1E** (original Fig. 1E to **new Fig. 1F**). We have replaced Fig. 2A with clearer images of the same sample groups. We have now also included a quantification of the average size of the top 10 largest lysosomes for each cell in the sample groups (**new Fig. 2C**), which showed a similar trend as the quantification of total cellular lysosomes (**new Fig. 2B**).

Fig. 2B

Fig. 2C

3. Lysotracker quantitation shown in Supp Fig2 should be in the main figure2 and representative images should be shown for Supp Fig2C.

Response: We have moved the quantification together with a new plot quantifying largest lysosomes for cells in each sample group in the main figure (new Fig. 2B & 2C). Representative images for Fig. S2C (new Fig. S2B) were added as new Fig. S2C.

Fig. S2B

Fig. S2C

4. Data supporting purity of "purified lysosomes" used in Figure 2C should be shown in Supp Fig2. Are they positive for other endosomal markers (ie, early v late endosome markers, eg Rab5/EEA1 v RAB7/LBPA)? What about other SNXs or known SNX binding proteins? Lipid droplet markers (see below)?

Response: We have performed Western blot analyses on purified lysosomes with antibodies against SNX2, SNX5, SNX8, SNX9, and markers of early endosome (EEA1), late endosome and lysosome (RAB7 and LAMP1), Golgi apparatus (GM130), liposome (Perilipin-2), and mitochondria (Complex-II).

As analyses of results shown in **new Fig. S2E**, the purified portion contained mostly late endosomes and lysosomes, while early endosomes, Golgi apparatus and mitochondria were absent from the purified fraction. A portion of perilipin 2 was found in the purified fraction, indicating that lipid droplet may be present in the purified fraction. Interestingly, all sorting nexins examined were found in the purified fraction, which is likely because some of these sorting nexins, though may not participate in lysosome tubulation, are present on late endosomes for processes like retrograde transport ([1, 2]).

Fig. S2E

5. Fig 2E, the control lysosomes look more like lipid droplets and are there any lysosomes in the +SNX8 panel?

Response: The original Fig. 2E is now Fig. 2F. To examine possible contamination from lipid droplets, we performed Western blot analysis on purified lysosomes with the lipid droplet marker Perilipin-2. Results suggest that some lipid droplet did pass through to the final lysosomal fraction. For the +SNX8 panel, lysosomes were still present and many were with tubular structures, similar to those shown in Fig. S6E. We have replaced the image with another image clearly showing a tubule stretching from the lysosome (**new Fig. 2F**).

Fig. 2F

Clathrin has been implicated in budding from lysosomes and there could be a hint of a clathrin lattice on the +SNX8 mutant lysosome?

Response: We examined possible involvement of Clathrin per the Reviewer's suggestion.

- ① Through stacking multiple frames capturing the same tubule, we did not identify discernible Clathrin lattice structures on SNX8-driven *in vitro* tubules (**rebuttal Fig. 7**).
- ② Western blot analysis showed that Clathrin was not detectable in the purified lysosomal fraction (**rebuttal Fig. 8**).

Therefore, our results suggest that Clathrin is not required for tubule formation in our *in vitro* tubulation assays. We are aware that 10 μ M SNX8 used in our assay is well above physiological ranges. Therefore, whether Clathrin is required for SNX8-driven tubule formation *in vivo* remains to be solved with further studies.

Rebuttal Fig. 7

Rebuttal Fig. 8

6. Fig 3D shows a reduction in transport to the Golgi but it is impossible to say if lysosome to Golgi transport is disrupted without showing a lysosomal marker as well.

Response: We thank the Reviewer for the comment and have repeated BODIPY-LacCer staining with the expression of LAMP1-mCherry. Results suggest that most dispersive puncta we observed with BODIPY-LacCer were LAMP1-positive as well (*new Fig. 4D*), suggesting that these puncta were late-endosomal/lysosomal.

Fig. 4D

7. Fig 4D-F is not discussed in the text! 4D should have lysosomal and TGN markers co-stained and representative images/movies of tubule formation should accompany 4E.

Response: We thank the Reviewer for pointing this out, we have now added description of these results in the text. Co-staining of LAMP1 and TGN markers in BODIPY-LacCer assay is difficult as this involves three channels and the performance of BFP that we have is not satisfactory. However, we did perform BODIPY-LacCer assay in the presence of LAMP1 expression, as we reason that disruption of lysosome-to-Golgi retrograde transport is better presented with dispersive lysosomal BODIPY-LacCer puncta. We have also compared PBS vs. Elemicin for WT, GLA-KO, HEXA-KO, and NPC1-KO cells with BODIPY-LacCer staining (*rebuttal Fig. 4* on rebuttal page 4, *new Fig. 6E*), which showed similar rescue effects to SNX8 overexpression. We have replaced the original Fig. 4D and Fig. S4C with new data (*new Fig. 4D* above, *new Fig. S4C*). For Fig. 4E, it was replaced by a new plot with stably expressed GFP-SNX8 rather than transient expression (*new Fig. 4E*), and representative images were now added to the *new Fig. S4D* (see above).

8. From the images shown in 5A, it can't be determined if lysosomes are enlarged. Increased lysotracker signal could be due to increased lysosome numbers/clustering or even reduced pH.

Response: We agree with the Reviewer that the magnification of these images was not enough to clearly visualize lysosome size. We have now provided insets with larger magnification to clearly visualize single lysosomes (new Fig. 5A), and have also included a new quantification (new Fig. S4E) of larger lysosomes in these cells similar to new Fig. 2C.

9. Supp Fig 6A shows some very nice blots consistent with increased SNX8-lysosome interaction following treatment with drugs, but a no lysosomes control should be included to show that the drugs aren't precipitating SNX8 in the pellet.

Response: We like to politely point out that a no lysosomes control was already included in the blot (1st lane), as well as a lysosomes-only control (last lane), the blot legend may have been unclear on that point. We have modified the blot legend to make it clearer. We thank the Reviewer for pointing this out.

Fig. S6A

It would be nice to see IPs of endogenous protein from whole cells and/or immunofluorescence of SNX8/lysotracker \pm the three molecules.

Response: We thank the Reviewer for this great suggestion. We treated WT and NPC1-KO cells with Elemicin, which we focused on in our mouse experiments, and purified lysosomes from these cells and analyzed lysosomal SNX8 (new Fig. S6D). Results showed that application of Elemicin increased binding of SNX8 to lysosomes in cells within 10-30 minutes, which was in accordance with our *in vitro* binding assays. This result further strengthened our conclusion that Elemicin enhanced lysosomal functions through increased binding of SNX8 to lysosomes.

Fig. S6D

10. Supp Fig 6C, there appears to be a dose-dependent reduction in SNX8 with elemicin - it might be worth checking effects of treating with lower concentrations (eg 10uM).

Response: We thank the Reviewer for the suggestion, we have repeated this experiment with lower concentrations of drugs at 0.1, 1, and 10 μM (Rebuttal Fig. 9), and results showed that the expression level of SNX8 remained unaffected. As the point of this panel is to show that these drugs did not alter SNX8 expression, we feel that it is better to keep the blot with higher drug concentrations, and thus did not replace the original blot.

Rebuttal Fig. 9

Minor points:

- *Figure 1, the rationale for the starve condition should be explained in the text.*

Response: We thank the Reviewer for the point. Lysosome reformation was first characterized in 2010, and prolonged starvation is one of the canonical conditions to trigger lysosome reformation, which needs lysosome tubulation as a platform. We added an explanation for the rationale of the starvation condition in the main text.

- *p7, line 21, I think should read SNX8 could bind PI(5)P and PI(3,5)P2.*

Response: We have made the correction.

Reviewer #2

In this manuscript, the authors propose that the sorting nexin SNX8 is required for lysosome reformation, and that enhancing SNX8 function may be a potential therapeutic approach for treating lysosomal storage disorders. SNX8 knockout leads to lysosome swelling, reduced lysosome tubulation and cholesterol accumulation, which is restored by recombinant SNX8 expression. SNX8 overexpression, or small molecules that promote SNX8 function, reverse lysosome swelling in several LSD cellular models, and restore muscle function in a Hexb knockout mouse model of gangliosidosis. Overall, these findings are of significance to the cell biology field as lysosome reformation is an area of great interest. However, the physiological data is fairly limited, and it is challenging to understand how enhanced lysosome reformation would restore the defective lysosomal enzyme function of LSDs. Some aspects of the methodology for measuring lysosome reformation also need improvement, and it is not clear how SNX8 is recruited to lysosomes.

Major comments

1. Although LSDs caused by loss of lysosome enzyme function can exhibit reduced lysosome reformation, it is unclear how SNX8-dependent lysosome reformation would restore lysosome function and improve disease pathogenesis as newly formed lysosomes would also be expected to retain defective enzyme function.

Response: We thank the Reviewer for raising this interesting point. Lysosomes are dynamic organelles and their functions rely heavily on trafficking and turnover. Majority of LSD phenotypes are actually related to secondary storage (e.g. cholesterol) and trafficking defects rather than the primary defect of the mutated gene. TFEB overexpression was reported to suppress phenotypes of lysosome-related diseases [3-6], so it is not totally surprising to see that SNX8-mediated lysosome reformation is able to replenish the pool of functional lysosomes and facilitate lysosomal membrane trafficking/recycling, thereby alleviating secondary storage and membrane trafficking defects. We have now expanded discussion on this point in the discussion section.

This is likely to limit the clinical applications of SNX8 as a therapeutic target. SNX8 activation appears to acutely alleviate GM2 accumulation and motor function in vivo, but it is difficult to determine from the data presented here whether disease is reversed as there is no evidence of impact on long term histopathology, motor function or survival of LSDs.

Response: We agree with the Reviewer that long-term effects of SNX8 activation on LSD phenotypes in vivo, either by ectopic expression or by small molecules, should be assessed before consolidating SNX8 as a valid therapeutic target, and we are happy to take the Reviewer's suggestions to perform these assessments. However, due to the low birth rate of LSD homozygous mice (both Hexb^{-/-} and NPC1^{-/-} mice birth rate from our heterozygous mating parents were recorded to be around 1% of the offspring), the time required to make these assessments would take beyond reasonable revision time. Thus we plan to include these assessments in future investigations.

Similarly, cell line rescue experiments are performed by transient transfection of GFP-SNX8, does stable GFP-SNX8 expression elicit the same sustained effect?

Response: We have generated pooled GFP-SNX8/SNX8 stable cells with WT, GLA-KO, HEXA-KO, NPC1-KO cells, and replaced overexpression experiments shown in Fig. 4C-4F with GFP-SNX8 (Fig. 4C, E, F) or SNX8 (Fig. 4D) stable cells. Results of these experiments were on the same trend with overexpression, suggesting that the rescue effects of SNX8 overexpression on LSD phenotypes is long-lasting and genuine.

Fig. 4C-F

2. Much of the authors' data relies on changes in the intensity and/or size of lysotracker, a dye that labels acidic cellular compartments and is commonly used to estimate total lysosome volume. However, as defective lysosome reformation leads to not only enlarged lysosomes but also reduced lysosome numbers (PMID: 20526321, 28857423, 33119550),

the authors also need to determine whether there is a corresponding change in lysosome numbers with SNX8 KO or overexpression, or with SNX8-activating small molecules. As lysotracker does not label terminal storage lysosomes (ie nascent lysosomes generated by reformation) (PMID: 28857423, 20526321, 27498570), this analysis should be performed with LAMP1 or LAMP2 antibody staining to label all lysosomes.

Response: We thank and agree with the Reviewer for his/her valuable comment. We chose lysotracker in some experiments because lysotracker intensity can be quickly read with flow cytometry and is a good indicator of total lysosome volumes (provided that the pH is not significantly altered, though).

Per the Reviewer's comment, we performed quantifications of the diameter of top 10 largest lysosomes in cells for WT, SNX8-KO cells, SNX2/8-DKO+SNX8 overexpression (new Fig. S2B), and WT/NPC1-KO cell +PBS or +Elemicin with LAMP1 staining (Rebuttal Fig. 10, next page). We tried to quantify cellular number of lysosomes but found out that it is too difficult to quantify, as lysosomes are often clustered close to the MTOC, and it is very hard to distinguish very small lysosomes from random noise, too. Provided that total volume of lysosomes is a function of lysosome size and number, we feel that a combination of total lysosome volume and an estimation of lysosome size can provide enough insights into the trend of lysosome size/number changes in these cells. Our results showed that SNX8-KO and NPC1-KO both induced significant increase in the volume of larger lysosomes, while overexpression of SNX8 (new Fig. S2B) or Elemicin treatment (Rebuttal Fig. 10, next page) can reduce the volume of large lysosomes.

Rebuttal Fig. 10

3. As the authors acknowledge, cells undergo lysosome reformation from several hybrid organelles (eg autolysosomes, endolysosomes) in response to different stimuli, and distinct molecular machinery facilitate reformation from these different compartments (PMID: 33950241). Which hybrid organelles does SNX8 regulate tubulation from, and is this required for the lysosomal degradation of particular endocytic and/or autophagic cargoes? Is this function nutrient dependent or independent?

Response: We thank the Reviewer for the comments/questions.

- ① We have assessed the co-localization of SNX8-positive vesicles with LC3-mCherry (autolysosomes) and TR-dextran (endolysosomes). Imaging results (Rebuttal Fig. 11) showed that both markers partially co-localized with SNX8-positive vesicles, suggesting that SNX8 binds to both autolysosomes and endolysosomes and thus is not selective for the type of hybrid organelle.

Rebuttal Fig. 11

- ② We have assessed SNX8-regulated tubulation under complete medium and serum starvation in WT cells with SNX8-GFP overexpression. Results showed that although SNX8 overexpression was able to drive basal tubulation, nutrient starvation significantly increased the tubulation process (Rebuttal Fig. 12).

Rebuttal Fig. 12

4. The authors claim that SNX8 is recruited to lysosomes through its interaction with PI(5)P and/or PI(3,5)P₂, but they provide little evidence for this.

The authors show that mutating the K135 residue reduces PI(3,5)P₂ and PI(5)P binding in vitro, but this mutant generates thinner, distorted tubules of purified lysosomes, suggesting this mutation may instead affect SNX8 membrane deformation function. What is the rationale for mutating this particular residue?

The authors need to determine whether this mutation affects SNX8 localization to lysosomes, and whether interfering with PI(3,5)P₂ or PI(5)P levels affects wild-type SNX8 localization. The authors should also examine whether expression of the K135A mutant in SNX8 KO restores lysosome number and size.

Response: We thank the reviewer and have done the following to address his/her concerns.

- ① We performed some experiments with the SNX8-K135A mutant (Rebuttal Fig. 13, please see below for detailed discussion). We may require much figure space to explain the observed results and thus, we removed those data from the manuscript for more focused and straightforward presentation of our work.

Rebuttal Fig. 13

- ② Rationale for K135A mutation: K135 is the key phosphoinositide binding residue in the PX domain of SNX8, determined through domain comparison with PX domains of other SNXs, and mutation of this residue to Alanine was shown to abolish phosphoinositide binding for other SNXs [7]. We have now made this rationale clearer in the main text.
- ③ The Reviewer pointed out a very interesting phenomenon for SNX8-K135A in the *in vitro* tubulation assay, which we also noticed. Our data and previous reports [7, 8] suggest that the function of the PX domain is to locate sorting nexins to membranes to create a high local concentration to initiate membrane curvature, which is done by the BAR domain. Therefore, the high concentration of SNX8 protein in *in vitro* assays may waive the requirement of PX domain-mediated guiding to the lysosomal surface, explaining why SNX8-K135A is able to generate tubular structures. For the distorted morphology, we currently have no answer, but guess that this maybe a result of the PX domain of SNX8-K135A not associated with the membrane, but this will require future studies to solve possible differences between WT SNX8 and SNX8-K135A superstructures.
- ④ We have performed analysis on the size of lysosomes in SNX8-KO cells expressing SNX8-K135A in comparison with the GFP control and WT SNX8. Interestingly, results showed that although the SNX8-K135A mutant showed an almost diffused cytosolic distribution close to that of GFP, overexpression of this mutant was still able to rescue the enlarged lysosome phenotype, though less effectively than WT SNX8 (see previous page for Rebuttal Fig. 13). This is likely due to the fact that SNX8-K135A mutant was still able to drive tubulation under high concentration. When overexpressed, SNX8-K135A is not actively recruited to lysosomal membrane but the high concentration was likely still able to drive tubulation at lower efficiency.
- ⑤ We have done another experiment with apilimod, a potent PIKfyve inhibitor that suppresses PI(3,5)P2 synthesis, and results in vastly enlarged endocytic vesicles [9, 10].

Under apilimod treatment, we stained LAMP1 in cells overexpressing SNX8-GFP, or co-stained LAMP1 and endogenous SNX8. Clearly the treatment abolished the localization of endogenous SNX8 to *LAMP1-positive* vesicles; Interestingly, overexpressed SNX8-GFP still localized significantly to LAMP1-positive vesicles (see previous page for Rebuttal Fig. 13). We also observed that overexpression of SNX8-GFP significantly suppressed the enlargement of LAMP1-positive vesicles, suggesting that overexpressed SNX8 actively reduced lysosome size in response to apilimod treatment. As the rationale proposed here is somewhat less straight forward, we chose not to include this data in the final manuscript for better delivery of major conclusions.

5. The data showing that SNX8 loss enhances cell death with repeated serum starvation is not particularly convincing, as very little cell death seems to occur under these conditions (only 2% of cells are dead in starved SNX8 KO sample). The authors should instead perform starvation with glutamine-free DMEM or EBSS/HBSS treatment, which may induce more widespread cell death (PMID: 26139536, 35968799).

Response: We thank the Reviewer for the great suggestion and have performed starvation assays using glutamine-free DMEM and HBSS. However, we would like to politely point out that in our original submission, we have already included starvation data using a strong starvation buffer (STB, Fig. 3E) that contains 1% BSA, 140 mM NaCl, 1 mM CaCl₂, 1 mM MgCl₂, 5 mM Glucose, 20 mM HEPES at pH 7.4. Under this starvation condition, cells were not able to withstand repeated starvation, so we simply performed 6 h starvation, and results showed around 20% starvation-induced cell death in SNX8-KO cells compared to less than 5% starvation-induced cell death in WT cells. Similar to STB buffer, HBSS starvation was also too severe for repeated starvation, and we adopted a single dose of starvation at 8 h, while for glutamine-free DMEM, we adopted the same repeated starvation protocol as DMEM starvation (new Fig. 3F and Fig. S3D). HBSS starvation showed similar trend as STB starvation, while glutamine-free DMEM induced significantly higher number of cell death than normal DMEM. In both cases, SNX8-KO cells showed significantly higher death rate than WT cells. We replaced the original DMEM-repeated starvation data with the new Gln-free-DMEM data, and we have also replaced Fig. 4F with a new Fig. 4F with Gln-free-DMEM and SNX8-stable cells.

6. There is insufficient evidence to conclude whether elemicin, isopsoralen or morroniside enhance SNX8 localization to lysosomes. In Figure 7A, there are no loading controls to show whether there is equal lysosome input.

Response: We thank the Reviewer for the comment. According to the content, we suppose the Reviewer is referring to Fig. S6A here. We did not include LAMP1 blots in the original manuscript because each of the drug group had its own control, and putting them all on seemed somewhat crowded. We have now included these blots in the **new Fig. S6A**.

There is no statistical analysis of the graph in Figure 7B, and the legend does not state what the quantification was normalized to.

Response: According to the comment, we suppose the Reviewer is referring to Fig. S6B here. We have now performed statistical analysis on this plot (**new Fig. S6B**). We did make a mistake in the legend for Fig. S6B and we thank the Reviewer for pointing this out. This is now corrected.

Changes in SNX8 localization to lysosomes should also be confirmed by immunofluorescence experiments.

Response: We agree with the Reviewer that changes in lysosomal SNX8 in response to drug treatment should be assessed. We feel that immunofluorescence is not the best for such purpose as fluorescence intensity between each sample cannot be convincingly compared. Instead, we chose to perform Western blots against SNX8 with lysosomes purified from WT and NPC1-KO cells with or without Elemicin treatment (see previous page for **new Fig. S6D**, see also **response to Reviewer #1, point 9**). Results clearly showed that SNX8 binding to lysosomes is increased under 10 μM Elemicin treatment.

The authors should also determine whether these drugs affect SNX8 function at endosomal compartments, as has been previously described (eg PMID 19782049, 34524084).

Response: We thank the Reviewer for the suggestion. Previous reports (PMID 19782049, 34524084, 21973056) showed that SNX8 promotes endosomal recycling in yeast and tubulation in mammalian cells, and promotes retrograde delivery of endocytic cargo to the Golgi apparatus.

In the original manuscript, we showed that small molecule drugs enhanced LAMP1-positive tubulation (original Fig. 6H). We have now added experiments on the effect of Elemicin on retrograde transport (**new Fig. 6E**, see also **Rebuttal Fig. 4 in response to Reviewer 1**), results of which suggest that Elemicin was able to enhance retrograde transport of endocytosed Lactosylceramide to the Golgi apparatus in LSD cells, suggesting that Elemicin does at least affect late-endosomal functions of SNX8. As dissection of compartmental functions of SNX8 was not a focus of this study, we chose not to pursue further on this branch.

7. Some further in vivo characterization is required. Were the behavioural studies (grip strength and rotarod experiments) performed by investigators blinded to treatments? This is critical given the potential for experimenter bias when performing rodent behavioural-type experiments.

Response: The experimenters (co-authors CL and AW) were blind to the animal groups.

Does recombinant SNX8 expression or drug treatment also restore the level of apoptosis, lysosome homeostasis or brain histopathology as previously reported for HexB KO mice (PMID: 9302266, 7550345, 8789434)? The authors would also need to show that recombinant SNX8 expression or drug treatment restores lysosome numbers in vivo.

Response: We have performed staining of LAMP1 and cleaved Caspase-3 in WT and Hexb^{-/-} mouse brain sections with either Elemicin treatment or SNX8 ectopic expression (**Rebuttal Fig. 14**, next page). At 3 months of age, although cell density in Hexb^{-/-} brains was apparently lower than in WT brains, the level of cleaved Caspase-3 staining was low in both, while Hexb^{-/-} brains showed a small increase in Caspase-3 positive population (no statistical significance, though), suggesting that the reduction in brain cell population was the result of mild but long-lasting cell loss instead of sudden cell death surges.

Both Elemicin and SNX8 overexpression reversed the increase in Caspase-3 staining and significantly restored cell population in Hexb^{-/-} brains. For Lamp1, as staining would require sectioning mouse tissue in thin slices, it was not feasible for us to quantify the number of lysosomes in cells. We instead quantified total Lamp1 staining, and found that Elemicin or SNX8 overexpression both suppressed the increase in Lamp1 staining in Hexb^{-/-} brains.

Rebuttal Fig. 14

In control studies, GFP tissue staining should be performed to show the proportion of cells expressing GFP or GFP-SNX8, and confirm the specificity of GFP-SNX8 expression in brain versus other tissues.

Response: We performed GFP staining in both WT and Hexb^{-/-} mouse brains expressing ectopic GFP or GFP-SNX8 (**Rebuttal Fig. 15**). Images clearly showed diffused GFP signal in most regions of the brain. We did not confirm if GFP-SNX8 is leaky to other tissues, as our current goal of the study is to confirm whether SNX8 expression is effective in rescuing brain defects in Hexb^{-/-} mice. Determination of the degree of importance of SNX8 expression in different tissues would be a very interesting aspect in our follow-up studies.

Rebuttal Fig. 15

Minor comments

1. Another group recently reported that SNX2 is recruited to endolysosomes by PI(3,5)P2 and promotes lysosome reformation (PMID: 35968799). The authors should acknowledge this study and speculate as to why they observe more modest effects with SNX2 KO in their cellular models.

Response: We thank the Reviewer for the kind reminder. The paper the Reviewer mentioned (entitled “Endosome maturation links PI3K α signaling to lysosome repopulation during basal autophagy”) describes a PI(3,4)P2-4-phosphatase INPP4B that regulates PI3P production on endolysosomes, and lysosome numbers, and they report that SNX2 participates in this regulation by promoting lysosome fission.

However, we feel that the authors of this paper were in fact describing ordinary lysosomal membrane fission instead of lysosome reformation, and there are two main reasons to support our view:

- ① Lysosome reformation is an on-demand process that happens on consumed lysosomes [11-14], but the authors of the PMID: 35968799 paper have not assessed any conditions known to elicit lysosome reformation (e.g. prolonged starvation or phagocytosis) but instead used PIKfyve inhibitor washout experiments.
- ② Lysosome reformation requires a tubular platform, while ordinary lysosomal fission

events (including retrograde transport vesicle fission) bud directly from lysosomes without the formation of a tubular structure, and all data in the paper were showing membrane fission without any tubular structure, thus were not lysosome reformation but other types of direct membrane fission from the vesicle, likely including retrograde transport, in which SNX2 is well known to take part. In their data, depletion of SNX2 resulted in marginal decrease in lysosome number (Fig. 6B of the paper, $p=0.049$) in MCF-7 cells, while lysosome size and total lysosome volume were not assessed.

In our hands, SNX2-KO did not result in significant changes in total lysosome volume nor an increase in the diameter of large lysosomes in HeLa cells (new Fig. S2B, S2C), the differences of the effect of SNX2 depletion may have been an issue of different cell types. We would like to point out, though, in our experiments, SNX2/8-DKO cells showed in general more severe phenotypes than SNX8-KO cells, including lysosome tubulation and lysosome volume, suggesting that SNX2 may serve as a redundancy for SNX8 in lysosome tubulation/reformation processes. We have now included this paper with related discussions in the discussion section in the revised main text.

Fig. S2B

Fig. S2C

2. In Figure 1D, separated LAMP1 and SNX2/8 channels should also be included as LAMP1-positive tubules are difficult to see in the merged images.

Response: Single channels of these images were now shown in new Fig. S1C per the Reviewer's suggestion.

Fig. S1C

3. Figure S1E should be moved to main figures with the corresponding quantification (Figure 1E). A GFP-vector control should also be included, as plasmid transfection may effect lysosome homeostasis. GFP-SNX8 expression does not appear to restore the swollen lysosome size observed in SNX8 or SNX2/8 KO cells.

Response: we thank the Reviewer for the comment! We have now moved the original Fig. S1E to the main figure (new Fig. 1E). For the swollen lysosome size phenotype, we chose these images to better present tubule formation without paying attention to lysosome sizes. We have re-examined our data and have now replaced with images that are representative of both tubulation and lysosome size for each group.

Fig. 1E

4. In Figure 2B, there are multiple SNX2 bands per lane and some are present in the IgG control.

Response: The top band is non-specific, probably caused by nominal cross-reactivity of the IgG in the IP sample. We have now added a plot legend to indicate it.

5. Images of starved cells in Figure S2B should also be included.

Response: We figure that the Reviewer is referring to Fig. S2C here. Representative images were now shown as **new Fig. S2C**.

Fig. S2C

6. In Figure S2C, statistical comparison between each cell line at the 6 hour timepoint needs to be performed to draw appropriate conclusions.

Response: We thank the Reviewer for the reminder and have now included these statistical comparisons in the **new Fig. S2B**.

Fig. S2B

7. How did the authors quantify the lysosome-specific pool of filipin versus other cellular pools? Lysosome co-labeling should be shown for all the lysosomal filipin experiments (Figure 3B, 4C, 6C).

Response: With our experience and co-localization experiments with LAMP1 ([9] and **Rebuttal Fig. 16**), cytosolic Filipin puncta are mostly, if not all, lysosomal. However, quantification using LAMP1 signal as a mask is much more time consuming and also introduces a much more variable factor – the difference in the expression level of LAMP1 in each cell, thus we chose to quantify Filipin staining without using LAMP1 as a reference. We feel that as long as most cytosolic cholesterol storage is lysosomal, quantifications in the current form are suitable for us to draw conclusions on cholesterol storage levels.

Rebuttal Fig. 16

8. On page 19, line 18, should “50 L” be “50 μ L”?

Response: We thank the Reviewer for pointing this out, this was a font error introduced by Word format conversion. The error has been corrected. We would appreciate if the Editor can proofread Greek symbols, especially “ μ m”, “ μ M”, and “ μ L”, in case that format

conversion errors occur again, thanks a lot!

9. For lysosome isolation experiments, how did the authors confirm there were no other contaminating organelles present?

Response: We have now inserted Western blots against organellar markers for the lysosome purification fractions (**new Fig. S2E**).

Fig. S2E

10. All figures should show individual data points, and legends should state the number of cells or mice analyzed.

Response: We have modified our plots and legends accordingly.

11. Many graphs have axis breaks where inappropriate (Figure 1B, 5G, 7E, S3B).

Response: These breaks have been removed.

12. Immunofluorescence and immunoblotting protocols should be properly described in methods section.

Response: We have re-written the Methods section to better clarify the protocols.

Reviewer #3

The manuscript by Xinran Li and colleagues reports that SNX8 – and to a lesser extent SNX2 – has a role in tubulation of lysosomal membrane when overexpressed. SNX8 is one of several PX-BAR family members proteins with known roles in tubulation of membranes in the endomembrane system. Loss of SNX8 (by knockout) leads to an enlarged lysosomal compartment and several trafficking and localization defects, consistent with those seen in cell models of lysosomal storage disorders. The authors point out one avenue for targeting lysosomal storage disorders may be improving lysosome reformation, which is known to require membrane tubulation. The authors show that overexpression of SNX8 reversed some of the storage-disorder-like phenotypes in cultured cells as well as in brains of a mouse model of a lysosomal storage disorder. The authors propose that improving lysosome reformation by, for example, augmenting the function of SNX8 (shown in this work by overexpression of SNX8) may be an approach that can be used to target and treat several lysosomal storage disorders. The authors finish this work by identifying 3 natural products that enhance SNX8 association with lysosomal membranes and reversed storage disorder phenotypes in cultured cells and their mouse model. These products did not alter SNX8 expression.

Overall, the central hypothesis is attractive and straightforward, and the authors present some intriguing data suggesting that strategies to enhance SNX8 tubulation function may aid in targeting storage disorders. However, there are a number of shortcomings.

The authors show that overexpression in COS7 cells leads to some colocalization with the standard lysosomal marker LAMP1. The authors test only SNX-BAR family members, most of which have previously been described to be associated with the endosomal compartment. Clearly, overexpression may lead to significant changes in the steady state distribution of SNX2 and SNX8. The contrast with endogenous SNX2 and SNX8 is clear: the images shown in 1C show essentially no (SNX2) and perhaps some (at best) colocalization with SNX8. The authors may wish to conduct a more complete assessment and quantification of these proteins with EEA1 and certainly determine the correlation coefficients of endogenous SNX2 and SNX8 with LAMP to convey the changes brought on by overexpression.

Response: We agree with the Reviewer that overexpression does not draw definite conclusion on the localization of a protein, as non-biological levels of a protein may shift its localization to compartments it normally does not reside. We would like to point out, though, that we utilized this co-localization experiment as a start-off clue for our study, and further knock-out studies demonstrated that SNX8 indeed participates in lysosomal membrane trafficking. In order to address the Reviewer's comment, we performed co-immunostaining on endogenous SNX2/SNX8 with EEA1 and LAMP1 (**Rebuttal Fig. 17**, next page). Results showed that indeed, endogenous SNX2 only showed limited co-localization with LAMP1, while showing a significant co-localization with EEA1, suggesting that endogenous SNX2 is mostly early-endocytic when membrane associated. On the contrary, endogenous SNX8 showed significant co-localization with both EEA1 and LAMP1, suggesting that SNX8 is able to localize to both early- and late-endocytic membranes.

Rebuttal Fig. 17

A more complete characterization of this type may be of interest or necessary in view of the recent study (Suzuki and colleagues, 2021) demonstrating that highly expressed SNX8 in HeLa cells resulted in tubulation from endosomes. In the authors' system, is the endosomal compartment still distinct on SNX8 overexpression?

Response: We performed immunostaining of EEA1 in SNX8-overexpressing cells and analyzed the effect of SNX8 on the morphology of EEA1-positive vesicles (Rebuttal Fig. 18, next page). Results showed that the morphology of EEA1-positive vesicles were not obviously affected by SNX8 overexpression.

Rebuttal Fig. 18

The negative stain electron microscopy shown in Fig. 2E is uninterpretable in its current form. The authors use purified lysosomes and incubate it with purified SNX8 or a point mutant. It is not clear from the figure, legend, or associated text whether the lysosomes are purified from SNX2 or 8 knockout cells or not – is endogenous SNX8 or SNX2 present?

Response: We used SNX8-KO cells for the lysosome purification in EM experiments with SNX8-lysosome interaction. We have now empathized this in the figure legend.

The role of the purified SNX8 in the tubulation is hard to assess in this context without a demonstration (immunogold?) that it indeed associates with the purified lysosomes.

Response: We acknowledge the concern of the Reviewer and agree that showing immunogold staining will be best to demonstrate the presence of SNX8 on lysosome tubules. Unfortunately, after preliminary tests with immunogold staining and discussion with our EM expert, we concluded that lysosome tubules could only be detected if lysosomes were fixed onto the carbon-coated grid immediately after the *in vitro* tubulation assay without having gone through the immunogold staining protocol, which inevitably disrupts the tubular structure. However, as we mentioned above, all lysosomes we used in the *in vitro* lysosome tubulation assay were purified from SNX8-KO cells, ruling out possible interference from endogenous proteins. Plus, in Fig. S6, we showed that lysosomes purified from SNX8-KO cells did not respond to Elemicin and only very few tubules could be observed, while lysosomes from WT cells responded to both Elemicin treatment and SNX8 application to generate tubules. Therefore, we hope that the Reviewer would agree with us that SNX8 is indeed sufficient to drive lysosome tubulation.

Perhaps the authors could also show the quality of their SNX8 purification.

Response: We have performed Cossamie Blue staining of the leftover of the same batch of SNX8 protein used for the *in vitro* assays (Rebuttal Fig. 19, next page). Only a single band appeared in the staining corresponding to the size of SNX8, indicating no major contamination from other proteins in the purified SNX8, nor did the protein undergo significant degradation.

Rebuttal Fig. 19

Indeed, the result with the mutant K135A mutant, showing formation of thinner tubes than wild type, is unexpected. In other systems, mutations of phosphoinositide-binding residues in the PX domain blocks tubulation of model membranes entirely.

Response: The distorted tubules observed in the SNX8-K135A+lysosome sample is indeed interesting. We currently do not have data for a model, but our guess is that although the PX domain of SNX8-K135A mutant does not bind PI(3,5)P₂, the BAR domain is still intact and can interact with membrane and cause curvature. The high concentration of the protein we used in the experiment (10 µM) still guaranteed a high enough local concentration of SNX8 around lysosomal membrane to promote tubule formation. The distorted morphology may have been caused by the relief of the PX domain from the membrane, but this is purely putative, exact cause would require further structural studies.

The data with elemicin, isopsoralen and morroniside showing enhancement of SNX8 recruitment to lysosomes is interesting and perhaps a little surprising as the structures of the drugs shown in Sup Fig 5B are quite different. All stimulate binding of (HeLa endogenous) SNX8 and show increased binding, albeit to differing extents. It would be interesting to see whether SNX2 or other SNX proteins that don't rescue lysosomal storage disorder phenotypes or that don't colocalize with LAMP1 when overexpressed have this effect.

Response: We thank the Reviewer for the suggestion and have examined the interaction of SNX2, SNX5 and SNX9 with lysosomes with or without Elemicin. Results suggest that these sorting nexins don't respond to Elemicin (**Rebuttal Fig. 20**), thus the enhancement of lysosomal binding is specific to SNX8 at least among these sorting nexins.

Rebuttal Fig. 20

Smaller issues:

There are some significant factual inaccuracies in this work:

Page 5 Line 12: “The majority of the SNXs carry a BAR domain capable of inducing membrane curvature”. The majority of SNX proteins are not SNX-BARs and not all SNX-BARs can generate membrane curvature. The Collins lab has pointed out that the human genome has 49 PX domain-containing proteins (most are also known as SNX proteins) (Chandra and others, 2019) but only 12 are PX-BARs, or SNX-BARs (van Weering and others, 2010).

Response: We thank the Reviewer for the critique and we agree with the Reviewer. We have made corresponding corrections in the text.

Page 7 line 20: “SNX proteins are known to bind phosphoinositides through their PX domains”. This is not the case. The PX domains of SNX5/6 and SNX32 are cargo binding and don’t bind phosphoinositides (for example, Chandra and colleagues, 2019; Yong and colleagues, 2020)

Response: We thank the Reviewer for the kind reminder. The PX domains may not interact directly with phosphoinositides, but instead mediate protein-protein interactions which may in turn aid their membrane localization, although whether some PX domains mentioned here can bind to phosphoinositides is controversial [15-18]. We have now modified this sentence to avoid ambiguity or controversy, and we thank the Reviewer for pointing this out.

The data presented in Fig. 1B shows that SNX1 (not pursued) and SNX2 show an enhanced correlation coefficient with LAMP1 on overexpression. It is curious that their obligate dimerization partners (SNX 5 or 6) show lower correlation coefficients with LAMP1 than even the GFP control. Could the authors comment on this? This may be less a concern with SNX8 as it homodimerizes, but it indicates that there may be some challenges associated with overexpressing SNX-BARs.

Response: We thank the Reviewer for the question and discussion. When we first did the screen, we did not make any pre-assumptions and simply followed the co-localization results, and frankly speaking, did not pay attention to this question about SNX1/2 and SNX5/6 localization. Concerning this question, our view is that, in an overexpression system, the abundance of the protein is dozens of times higher than the endogenous level (we did some rough comparison on SNX8, which we estimated to overexpress about 30-50 folds higher than endogenous SNX8 with our expression vector). Therefore, concentrations of endogenous binding partners are too low to affect the localization of the majority of overexpressed SNXs, and their localization therefore is primarily determined by their intrinsic affinity towards the membrane. According to previous reports, SNX1/2 localize to both early and late endocytic compartments ([2, 18-20]), while SNX5 localizes primarily to early endosomes and similar vesicles such as macropinosomes ([17, 21]). We think this may explain the different localization results we see in Fig. S1A. For the comment on challenges with overexpression, please see our response at the beginning.

Page 9 line 12: “These cell lines all showed similar LSD phenotypes (Fig. 4 & Supplemental Fig. 4B, 4C)”. The lysosomal morphology of GLAKO, HEXAKO and NPC1KO cells are not

similar though they do all look increased compared to WT cells.

Response: We agree with the Reviewer and have modified the text accordingly.

1. Cullen, P.J. and H.C. Korswagen, *Sorting nexins provide diversity for retromer-dependent trafficking events*. Nat Cell Biol, 2011. **14**(1): p. 29-37.
2. Gullapalli, A., et al., *An essential role for SNX1 in lysosomal sorting of protease-activated receptor-1: evidence for retromer-, Hrs-, and Tsg101-independent functions of sorting nexins*. Mol Biol Cell, 2006. **17**(3): p. 1228-38.
3. Xu, J., X.Q. Zhang, and Z. Zhang, *Transcription factor EB agonists from natural products for treating human diseases with impaired autophagy-lysosome pathway*. Chin Med, 2020. **15**(1): p. 123.
4. Yang, C. and X. Wang, *Lysosome biogenesis: Regulation and functions*. J Cell Biol, 2021. **220**(6).
5. Tan, S., et al., *Pomegranate activates TFEB to promote autophagy-lysosomal fitness and mitophagy*. Sci Rep, 2019. **9**(1): p. 727.
6. Gatto, F., et al., *AAV-mediated transcription factor EB (TFEB) gene delivery ameliorates muscle pathology and function in the murine model of Pompe Disease*. Sci Rep, 2017. **7**(1): p. 15089.
7. Teasdale, R.D. and B.M. Collins, *Insights into the PX (phox-homology) domain and SNX (sorting nexin) protein families: structures, functions and roles in disease*. Biochem J, 2012. **441**(1): p. 39-59.
8. van Weering, J.R., P. Verkade, and P.J. Cullen, *SNX-BAR proteins in phosphoinositide-mediated, tubular-based endosomal sorting*. Semin Cell Dev Biol, 2010. **21**(4): p. 371-80.
9. Li, X., et al., *A molecular mechanism to regulate lysosome motility for lysosome positioning and tubulation*. Nat Cell Biol, 2016. **18**(4): p. 404-17.
10. Cai, X., et al., *PIKfyve, a class III PI kinase, is the target of the small molecular IL-12/IL-23 inhibitor apilimod and a player in Toll-like receptor signaling*. Chem Biol, 2013. **20**(7): p. 912-21.
11. Yu, L., et al., *Termination of autophagy and reformation of lysosomes regulated by mTOR*. Nature, 2010. **465**(7300): p. 942-6.
12. Chen, Y. and L. Yu, *Development of Research into Autophagic Lysosome Reformation*. Mol Cells, 2018. **41**(1): p. 45-49.
13. Pryor, P.R., et al., *The role of intraorganellar Ca(2+) in late endosome-lysosome heterotypic fusion and in the reformation of lysosomes from hybrid organelles*. J Cell Biol, 2000. **149**(5): p. 1053-62.
14. Mrakovic, A., et al., *Rab7 and Arl8 GTPases are necessary for lysosome tubulation in macrophages*. Traffic, 2012. **13**(12): p. 1667-79.
15. Chandra, M., et al., *Classification of the human phox homology (PX) domains based on their phosphoinositide binding specificities*. Nat Commun, 2019. **10**(1): p. 1528.
16. Niu, Y., et al., *PtdIns(4)P regulates retromer-motor interaction to facilitate dynein-cargo dissociation at the trans-Golgi network*. Nat Cell Biol, 2013. **15**(4): p. 417-29.
17. Merino-Trigo, A., et al., *Sorting nexin 5 is localized to a subdomain of the early endosomes and is recruited to the plasma membrane following EGF stimulation*. J Cell

- Sci, 2004. **117**(Pt 26): p. 6413-24.
18. Cozier, G.E., et al., *The phox homology (PX) domain-dependent, 3-phosphoinositide-mediated association of sorting nexin-1 with an early sorting endosomal compartment is required for its ability to regulate epidermal growth factor receptor degradation*. J Biol Chem, 2002. **277**(50): p. 48730-6.
 19. Kerr, M.C., et al., *Visualisation of macropinosome maturation by the recruitment of sorting nexins*. J Cell Sci, 2006. **119**(Pt 19): p. 3967-80.
 20. Rodgers, S.J., et al., *Endosome maturation links PI3Kalpha signaling to lysosome repopulation during basal autophagy*. EMBO J, 2022. **41**(19): p. e110398.
 21. Dong, X., et al., *Sorting nexin 5 mediates virus-induced autophagy and immunity*. Nature, 2021. **589**(7842): p. 456-461.

REVIEWER COMMENTS

Reviewer #1 (Remarks to the Author):

A substantial amount of work has been completed; the manuscript is improved as a result and the majority of my concerns have been addressed. Though I remain unconvinced that the long "tubular" structures seen by fluorescence are continuous tubules rather than "beads on a string" structures, since the data indicates a SNX8-dependent tubulation mechanism and importantly, an SNX8-dependent route for reversal of LSD phenotypes, I don't consider it necessary to investigate further (though this could potentially be done by loading lysosomes with HRP through fluid phase entry, inducing SNX8-dependent tubulation and imaging by conventional TEM using the DAB reaction product to visualise lysosome content/tubules).

It would also be nice to understand more about how tubulation/lysosome reformation might mediate clearance of cholesterol, eg in NPC when the cholesterol egress protein is missing - is the cholesterol redistributed (inc to ER for downregulation of SREBP) or removed (are lysosome tubules cholesterol-laden and are they recycled to the PM/exocytosed following fission?)? The question of how SNX8-induced tubulation reverses LSD phenotypes or what happens post-tubulation, ie what stops the newly formed protolysosomes from also becoming dysfunctional, isn't fully addressed but maybe beyond the scope of the study.

Minor comments:

- The difference between prolonged and overly sustained starvation would benefit from clarification. On p6, line 112 it says that lysosome reformation is triggered by prolonged starvation whereas on p7, line 132/3 says that overly sustained starvation suppresses lysosome reformation.
- Fig 2F please show lower mag SEM images, especially for the + SNX8 panel to make the whole lysosome readily visible (keeping higher mag insets). Can you show a more lysosome (less lipid droplet)-like example for ctrl lysosomes?
- Figure 4C and D are key experiments and need quantitation.
- Supplementary Figure 4D the upper pair of red arrows for the lower NPC panel are pointing to green staining that appears to be negative for Lamp1. Can you show an alternative image where the SNX8 staining is more convincingly emanating from a lysosome?
- Figure 6E, please show example representative images and clarify/demonstrate the meaning of "dispersive puncta".
- Figure 6G, the Y-axis is labelled "lysosomal" filipin. How is this determined? Is all filipin that isn't colocalised with lamp1 excluded by thresholding? Or does this refer to filipin within the PM (in which

case it is intracellular, not necessarily lysosomal) or total filipin (in which case it is cellular filipin, rather than lysosomal)?

Reviewer #2 (Remarks to the Author):

The authors have made significant efforts to address the criticisms of the manuscript, although some points require further clarification (see below). Once these are addressed, I would recommend the manuscript for publication.

1. It is necessary to include some of the rebuttal figures as supplementary figures in the manuscript for the authors to reasonably draw their conclusions, including rebuttal figures 10 (rescue of lysosome size with elemicin), 13 (effect of SNX8-K135A on lysosomes) and 15 (GFP-SNX8 staining in brain).

2. For new Fig S2B, authors state this is a measurement of “top 10 largest lysosomes” in the rebuttal letter, but this is labelled as “normalized total cellular lysotracker” in the manuscript. Can the authors clarify?

3. In the discussion, it is unclear what the authors mean when they speculate that SNX2 regulates “ordinary lysosome fission” as opposed to operating under “tubulation inducing conditions”. Presumably the authors mean that differences observed in SNX2 and SNX8 function may relate to their specific regulation of lysosome reformation under different nutrient conditions, but the functional redundancy between SNX proteins (such as that previously reported for SNX9/18 (PMID 20427313, 23823722)) may also contribute to this effect.

Reviewer #3 (Remarks to the Author):

I have carefully studied the revised manuscript by Li and colleagues on the role of SNX8 and, to a smaller extent, SNX2 in lysosome reformation. The authors have undertaken significant efforts to respond to my concerns as well as those of my colleagues who reviewed the first iteration of this work.

To summarize, the study shows that loss of SNX2 and SNX8 induces LSD-like phenotypes in cells, that could be suppressed by SNX8 overexpression. Further, overexpression of SNX8 in 3 HeLa lines with knockouts of genes responsible for different LSDs partially or significantly reverses the deficits seen in cellular models of LSDs, including increased lysosomal volume, cholesterol accumulation, and lysosome-

TGN traffic deficits, and decreased tolerance to starvation. This rescue extended to dermal fibroblasts from *Hexb*^{-/-} mice, where various perturbations were again decreased or reversed by SNX8 expression.

Some newly introduced work and the resulting figures have significantly strengthened the manuscript. From my perspective, the blots showing enhanced binding of SNX8 to lysosomes in the presence of elemicin, isopsoralen, and morroniside is a necessary addition. A concern about the 3 natural products generally improving association of any protein with a model membrane (in this case lysosomes) was tested in rebuttal fig. 20 for 3 other SNX-BAR proteins, although SNX8 was not included as a positive control. This blot is an important control for all the work on elemicin and perhaps merits inclusion in the final manuscript. Finally, in response to another reviewer, inclusion of the stable SNX8 expressing cell lines (Fig. 4C and D) overcomes some of the concerns related to transient overexpression.

My concern about the initial overexpression work with the various SNX-BARs does still persist. SNX1 and 2 may dimerize with either SNX5 or 6 but they cannot homodimerize. Expression of these alone is therefore problematic (even in the studies cited by the authors in their response to the reviewers). SNX8 only forms homodimers so this is not expected to lead to the same issues as overexpression of SNX2. However, all SNX2 overexpression results should therefore be taken with a caveat and should be mentioned in the manuscript (it cannot be ruled out, for example, that this may be a component of why SNX2 overexpression rescues LSD phenotypes less effectively than SNX8). I understand that the authors state that they screened overexpression of several SNX-BARs as the initial investigation in this work but for those that don't homodimerize the overexpression work be tough to interpret. The work with endogenous SNX2 and SNX8 (Fig. 1 C) is therefore a very important control for this work.

Overall, I feel that, while not all the nuances of SNX8 mechanism have been determined, the authors have demonstrated a clear role for SNX8 in lysosome reformation. The authors' proposal that targeting SNX8 function may improve lysosome reformation and so may be an avenue for tackling LSDs will be of interest to the community. I therefore continue to be enthusiastic about this work.

Smaller issues

The legend in Fig. 4C does not match the data – the images suggest a stable GFP-SNX-expressing line; the legend a transient transfection

The data presented in Fig 4B in the NPC1 KO line is a little puzzling. It's not clear SNX8 expression has any effect in this case.

Line 167: Fig. 3A-D does not refer to BODIPY-laccase – only 3D does

Line 845 – Lamp-mCherry

Line 850 – “Quantifications were shown in (H).” Presumably this refers to something else pre-reorganization

Sup Fig 2B: the annotation of starvation times is garbled.

Is Fig. 6I comparison of WT DMSO and Hexb^{-/-} + Elemicin 10 uM significant? Rescue appears partial?

Point-by-point reply to the reviewers' comments:

We thank the Editor and Reviewers for the enthusiasm for our work and the appreciation of our efforts for the revision. We also thank the Reviewers for their valuable suggestions and comments on the revised manuscript, which further improved the manuscript. We believe that we have addressed the remaining concerns from the Reviewers, and please see below for a point-by-point reply (with major changes marked up in **yellow highlights** in the manuscript file).

Reviewer #1 (Remarks to the Author):

A substantial amount of work has been completed; the manuscript is improved as a result and the majority of my concerns have been addressed. Though I remain unconvinced that the long "tubular" structures seen by fluorescence are continuous tubules rather than "beads on a string" structures, since the data indicates a SNX8-dependent tubulation mechanism and importantly, an SNX8-dependent route for reversal of LSD phenotypes, I don't consider it necessary to investigate further (though this could potentially be done by loading lysosomes with HRP through fluid phase entry, inducing SNX8-dependent tubulation and imaging by conventional TEM using the DAB reaction product to visualise lysosome content/tubules).

Response: we thank the Reviewer for the support of the manuscript! We also thank the Reviewer for providing a method for the EM detection of the tubular structure. We have collaborated with the EM facility on campus and tried several ways to visualize lysosomal tubules *in situ* but have unfortunately failed to acquire positive results within the revision period. We will continue to work on that and will also try the method provided by the Reviewer, and we hope that we will be able to acquire and show these results in our following studies.

It would also be nice to understand more about how tubulation/lysosome reformation might mediate clearance of cholesterol, eg in NPC when the cholesterol egress protein is missing - is the cholesterol redistributed (inc to ER for downregulation of SREBP) or removed (are lysosome tubules cholesterol-laden and are they recycled to the PM/exocytosed following fission)? The question of how SNX8-induced tubulation reverses LSD phenotypes or what happens post-tubulation, ie what stops the newly formed protolysosomes from also becoming dysfunctional, isn't fully addressed but maybe beyond the scope of the study.

Response: We agree with the Reviewer that the re-distribution of cholesterol is an interesting question for investigation. We did try to induce lysosome tubulation and monitor cholesterol distribution, but unfortunately, tubule structures were destructed after fixation protocol. On the other hand, we compared the non-puncta intracellular filipin intensity in NPC1-KO cells between SNX8-overexpressing cells and control cells. Interestingly, SNX8 overexpression

induced an increase of filipin intensity in some non-puncta, cloudy intracellular structures, indicating a re-distribution of cholesterol from lysosomes to these structures (Fig. R1). We speculate that these structures could be ER, and cholesterol may have re-distributed through retrograde transport enhanced by SNX8 overexpression. However, as the Reviewer also acknowledges, the detailed mechanism of the re-distribution of cholesterol is somewhat apart from the main story of the manuscript, so we chose not to include this part in the manuscript.

Fig. R1. Increased cholesterol in non-puncta intracellular structures in NPC1-KO cells overexpressing SNX8-GFP. Red arrows point out the non-puncta structures with high filipin signal, which is statistically significantly increased in cells overexpressing SNX8-GFP over the GFP-expressing control. Scale bar=10 μ m, **=p < 0.01.

Minor comments:

• *The difference between prolonged and overly sustained starvation would benefit from clarification. On p6, line 112 it says that lysosome reformation is triggered by prolonged starvation whereas on p7, line 132/3 says that overly sustained starvation suppresses lysosome reformation.*

Response: We apologize for the misunderstanding that our statements have

caused. The sentence on page 7, as the Reviewer mentioned, originally read “Since suppression of lysosome reformation, especially after overly sustained starvation, was reported to cause enlarged lysosomes, we monitored changes in lysosome volumes during starvation in these cells.” We actually meant to say that suppression of overly sustained starvation-induced lysosome reformation can lead to enlarged lysosomes. We have now re-arranged this sentence so that it now reads more clearly.

• *Fig 2F please show lower mag SEM images, especially for the + SNX8 panel to make the whole lysosome readily visible (keeping higher mag insets). Can you show a more lysosome (less lipid droplet)-like example for ctrl lysosomes?*

Response: We have replaced the images as the Reviewer requested.

• *Figure 4C and D are key experiments and need quantitation.*

Response: Quantifications of Fig. 4C was already included as Fig. S4B, and quantifications of Fig. 4D was included as Fig. S4C. They were not shown side-by-side due to space restrictions of the main figure, but were cited together in the main text. We have rephrased the main text citation of these figures for better presentation.

• *Supplementary Figure 4D the upper pair of red arrows for the lower NPC panel are pointing to green staining that appears to be negative for Lamp1. Can you show an alternative image where the SNX8 staining is more convincingly emanating from a lysosome?*

Response: Most tubulation images were snapshots from tubulation videos. As mCherry photobleaches faster than GFP, the red signal in videos drop quicker, and snapshots taken towards the end of videos tend to have lower lamp1-mCherry signal and therefore appear mostly green. As we explained in the last rebuttal, when we boost up intensity of the red channel, most “green” tubules still show mCherry signal on them. Nevertheless, we have replaced the NPC1-KO image with another image showing stronger lamp1-mCherry signal on tubules.

• *Figure 6E, please show example representative images and clarify/demonstrate the meaning of “dispersive puncta”.*

Response: We have now included representative images of Fig. 6E in Supplementary Fig. 6K, and we have also included a description of dispersive puncta and the quantification in the Methods section under “BODIPY-LacCer staining”, “For quantification of BODIPY-LacCer staining results, number of dispersive puncta BODIPY signal, which is scattered BODIPY dots more than 2 μ m away from the main perinuclear cluster, was counted for each cell. If no cluster is identifiable in the cell, then all BODIPY dots were counted.”

• *Figure 6G, the Y-axis is labelled “lysosomal” filipin. How is this determined? Is all filipin that isn’t colocalised with lamp1 excluded by thresholding? Or*

does this refer to filipin within the PM (in which case it is intracellular, not necessarily lysosomal) or total filipin (in which case it is cellular filipin, rather than lysosomal)?

Response: We showed in the previous rebuttal letter (Rebuttal Fig. 16, also shown as Fig. R2 below) that the intracellular puncta pool of filipin staining is, if not all, mostly Lamp1-colocalized. We have done a small-scale test using

Lamp1 signal as a mask and quantification in such a way yielded slightly smaller filipin intensity values than the “lysosomal” filipin intensity we quantified in the figures, but the filipin staining ratio remained about the same for all groups (Fig. R2). As making masks and quantify for all cells in this manner is too time consuming, we simply quantified intracellular filipin dots (by subtracting intracellular mean background intensity. We have now stated the whole quantification procedure in the Methods-Filipin staining section instead of citing our previous publication (Li, X. et al. 2016. A molecular mechanism to regulate lysosome motility for lysosome positioning and tubulation. *Nat Cell Biol* 18, 404-417.).

Fig. R2. Masking with Lamp1 or not does not significantly impact on the quantification of “lysosomal” filipin intensity. Samples from the same group were divided into two subgroups, one quantified using the method described in the manuscript, the other quantified using LAMP1 signal as a mask, and all data were normalized to WT cells quantified without Lamp1 masking.

Reviewer #2 (Remarks to the Author):

The authors have made significant efforts to address the criticisms of the manuscript, although some points require further clarification (see below). Once these are addressed, I would recommend the manuscript for publication.

Response: We thank the Reviewer for the support of the manuscript! We have made corrections and modifications per the Reviewer's comments, and hopefully the Reviewer's concerns have been addressed.

1. It is necessary to include some of the rebuttal figures as supplementary figures in the manuscript for the authors to reasonably draw their conclusions, including rebuttal figures 10 (rescue of lysosome size with elemicin), 13 (effect of SNX8-K135A on lysosomes) and 15 (GFP-SNX8 staining in brain).

Response: We thank the Reviewer for the suggestion! We have now included rebuttal figure 10 as **Fig. 6A, 6B**, the original Fig. 6A/B were now moved to supplementary and combined with the original Fig. S5C, S5D, S5E as new Fig. S6C, S6D. Rebuttal Fig. 13 is now added as **Fig. S2F, S2G**. Rebuttal Fig. 15 was inserted as **new Fig. S5B** (as a result, the original Fig. S5 was now Fig. S6).

2. For new Fig S2B, authors state this is a measurement of “top 10 largest lysosomes” in the rebuttal letter, but this is labelled as “normalized total cellular lysotracker” in the manuscript. Can the authors clarify?

Response: We thank the Reviewer for pointing this out and we are sorry for the confusion. We made a mistake in the Rebuttal letter: the description was for **Fig. 2C** but instead we mistakenly cited Fig. S2B. This was due to figure re-organization, and we forgot to modify the rebuttal letter accordingly. The main text and figures were correct and unaffected by this mistake.

3. In the discussion, it is unclear what the authors mean when they speculate that SNX2 regulates “ordinary lysosome fission” as opposed to operating under “tubulation inducing conditions”. Presumably the authors mean that differences observed in SNX2 and SNX8 function may relate to their specific regulation of lysosome reformation under different nutrient conditions, but the functional redundancy between SNX proteins (such as that previously reported for SNX9/18 (PMID 20427313, 23823722)) may also contribute to this effect.

Response: We have now expanded this paragraph in the discussion according to the Reviewer's comments, the part of the paragraph modified now reads “In the absence of tubulation-inducing conditions such as prolonged starvation or active phagocytosis, it is more likely that the reported study described normal lysosome fission events without the participation of lysosomal tubular structures, rather than induced lysosome tubulation. Nonetheless, further examinations across different cell types may be required to conclude the effect of SNX2 on lysosome tubulation/reformation, and our data on SNX2/8 DKO cells (**Fig. 3**) do indicate that SNX2 may serve as a functional redundant for SNX8 in lysosome tubulation.”

Reviewer #3 (Remarks to the Author):

I have carefully studied the revised manuscript by Li and colleagues on the role of SNX8 and, to a smaller extent, SNX2 in lysosome reformation. The authors have undertaken significant efforts to respond to my concerns as well as those of my colleagues who reviewed the first iteration of this work.

To summarize, the study shows that loss of SNX2 and SNX8 induces LSD-like phenotypes in cells, that could be suppressed by SNX8 overexpression. Further, overexpression of SNX8 in 3 HeLa lines with knockouts of genes responsible for different LSDs partially or significantly reverses the deficits seen cellular models of LSDs, including increased lysosomal volume, cholesterol accumulation, and lysosome-TGN traffic deficits, and decreased tolerance to starvation. This rescue extended to dermal fibroblasts from Hexb^{-/-} mice, where various perturbations were again decreased or reversed by SNX8 expression.

Some newly introduced work and the resulting figures have significantly strengthened the manuscript. From my perspective, the blots showing enhanced binding of SNX8 to lysosomes in the presence elemicin, isopsoralen, and morroniside is a necessary addition. A concern about the 3 natural products generally improving association of any protein with a model membrane (in this case lysosomes) was tested in rebuttal fig. 20 for 3 other SNX-BAR proteins, although SNX8 was not included as a positive control. This blot is an important control for all the work on elemicin and perhaps merits inclusion in the final manuscript. Finally, in response to another reviewer, inclusion of the stable SNX8 expressing cell lines (Fig. 4C and D) overcomes some of the concerns related to transient overexpression.

Response: We thank a lot for the Reviewer's support! Rebuttal Fig. 20 was now inserted to **new Fig. S7E** as suggested by the Reviewer, and the original Fig. S6E, S6F were now moved to new Fig. S7F, S7G (due to the inclusion of a new Fig. S5, the original Fig. S6 was now re-listed as Fig. S7).

My concern about the initial overexpression work with the various SNX-BARs does still persist. SNX1 and 2 may dimerize with either SNX5 or 6 but they cannot homodimerize. Expression of these alone is therefore problematic (even in the studies cited by the authors in their response to the reviewers). SNX8 only forms homodimers so this is not expected to lead to the same issues as overexpression of SNX2. However, all SNX2 overexpression results should therefore be taken with a caveat and should be mentioned in the manuscript (it cannot be ruled out, for example, that this may be a component of why SNX2 overexpression rescues LSD phenotypes less effectively than SNX8). I understand that the authors state that they screened overexpression of several SNX-BARs as the initial investigation in this work but for those that don't homodimerize the overexpression work be tough to interpret. The work with endogenous SNX2 and SNX8 (Fig. 1 C) is therefore a

very important control for this work.

Response: We thank the Reviewer for the comment and we agree with the Reviewer that our screen is not conclusive. We have now added a clarification at the end of the first paragraph of the Results section that reads “It is worth noting, however, that some sorting nexins heterodimerize, and therefore may require the co-expression of the binding partner to function properly, thus our screen does not entirely rule out possible participation of other sorting nexins in lysosome tubulation.”

Overall, I feel that, while not all the nuances of SNX8 mechanism have been determined, the authors have demonstrated a clear role for SNX8 in lysosome reformation. The authors’ proposal that targeting SNX8 function may improve lysosome reformation and so may be an avenue for tackling LSDs will be of interest to the community. I therefore continue to be enthusiastic about this work.

Response: Thanks again for the support of our work!

Smaller issues

The legend in Fig. 4C does not match the data - the images suggest a stable GFP-SNX-expressing line; the legend a transient transfection

Response: We thank the Reviewer for pointing out this mistake. The legend was copy-pasted and we forgot to modify along with changes in the figure content. We have now corrected the legend.

The data presented in Fig 4B in the NPC1 KO line is a little puzzling. It’s not clear SNX8 expression has any effect in this case.

Response: Fig. 4B shows the lysotracker intensity in WT or LSD model cell lines under control or a time course of starvation, and did not involve SNX8 overexpression. We did not include data on SNX8 overexpression’s effect on lysosome size in LSD cells. We have now included a new **Fig. S4B** that demonstrates the effect of SNX8 overexpression on lysotracker staining in LSD cells with flow cytometry.

Line 167: Fig. 3A-D does not refer to BODIPY-laccer - only 3D does

Response: We are sorry for the confusion. We have now separately cited different panels of Fig. 3 and Fig S3 in the main text.

Line 845 - Lamp-mCherry

Response: We have corrected this typo, thanks!

Line 850 - “Quantifications were shown in (H).” Presumably this refers to something else pre-reorganization

Response: Thanks for pointing this out! This was a leftover of the previous version and was now deleted.

Sup Fig 2B: the annotation of starvation times is garbled.

Response: The graph was corrected, thanks!

Is Fig. 6I comparison of WT DMSO and Hexb^{-/-} + Elemicin 10 uM significant? Rescue appears partial?

Response: The rescue was partial. We did not compare WT and rescue in the original version. We have now included statistical comparison between WT and Hexb^{-/-}+Elemicin.

REVIEWERS' COMMENTS

Reviewer #1 (Remarks to the Author):

All of my concerns have been addressed and I recommend the manuscript for publication - it will be of great interest to the community.

Reviewer #2 (Remarks to the Author):

The authors have adequately addressed all the comments and the manuscript is now suitable for publication.

Reviewer #3 (Remarks to the Author):

I have read the revised manuscript and feel that the authors have adequately addressed my concerns and those of my fellow reviewers. I support publication and look forward to seeing the study published.